# Selective inhibition of TGF-β1 produced by GARP-expressing Tregs overcomes resistance to PD-1/PD-L1 blockade in cancer

Grégoire de Streel [1], Charlotte Bertrand[1], Nicolas Chalon [1], Stéphanie Liénart[1], Orian Bricard [1], Sara Lecomte[1], Julien Devreux [1], Mélanie Gaignage[1], Gitte De Boeck[2], Lore Mariën [2], Inge Van De Walle [2], Bas van der Woning [2], Michael Saunders[2], Hans de Haard [2], Elien Vermeersch [3], Wim Maes[3], Hans Deckmyn [3], Pierre G. Coulie [1], Nicolas van Baren[1] & Sophie Lucas [1✉]

TGF-β1, β2 and β3 bind a common receptor to exert vastly diverse effects in cancer, supporting either tumor progression by favoring metastases and inhibiting anti-tumor immunity, or tumor suppression by inhibiting malignant cell proliferation. Global TGF-β inhibition thus bears the risk of undesired tumor-promoting effects. We show that selective blockade of TGF-β1 production by Tregs with antibodies against GARP:TGF-β1 complexes induces regressions of mouse tumors otherwise resistant to anti-PD-1 immunotherapy. Effects of combined GARP:TGF-β1/PD-1 blockade are immune-mediated, do not require FcγR-dependent functions and increase effector functions of anti-tumor CD8+ T cells without augmenting immune cell infiltration or depleting Tregs within tumors. We find GARP-expressing Tregs and evidence that they produce TGF-β1 in one third of human melanoma metastases. Our results suggest that anti-GARP:TGF-β1 mAbs, by selectively blocking a single TGF-β isoform emanating from a restricted cellular source exerting tumor-promoting activity, may overcome resistance to PD-1/PD-L1 blockade in patients with cancer.

[1] de Duve Institute, Université catholique de Louvain, 1200 Brussels, Belgium. [2] argenx, 9052 Zwijnaarde, Belgium. [3] Laboratory for Thrombosis Research, IRF Life Sciences, 8500 KU Leuven Campus Kulak Kortrijk, Kortrijk, Belgium. ✉email: sophie.lucas@uclouvain.be

Immunosuppression by regulatory T cells (Tregs) is indispensable to maintain peripheral immune tolerance, but is detrimental in cancer or chronic infections. Targeting Tregs or their functions in cancer patients has remained a coveted, but challenging and unmet therapeutic approach. Coveted, because notwithstanding the remarkable progress in cancer treatment achieved with monoclonal antibodies (mAbs) blocking the CTLA-4 or PD-1 inhibitory pathways, a vast majority of patients do not respond to immunotherapy due to primary or acquired resistance to T-cell-mediated anti-tumor immunity[1,2], and Tregs appear deleterious to anti-tumor immunity in most patients and cancer types[3–12]. Very recently, Tregs were even suggested to be amplified and contribute to disease hyperprogression in response to PD-1 blockade in a small subset of cancer patients[13]. Nevertheless, whereas mouse Tregs were shown to suppress immune responses by a variety of context-dependent mechanisms, which mechanism, if any, should be targeted to block suppression of anti-tumor immunity by Tregs in cancer patients is not known. None of the current cancer immunotherapies allows to specifically block Treg immunosuppression without killing these cells in the tumor microenvironment.

We recently identified a mechanism of immunosuppression by human Tregs that can be blocked by mAbs. This mechanism implicates production of the potently immunosuppressive TGF-β1 cytokine. Like most other immune cells, Tregs produce TGF-β1 in a latent, inactive form, in which the mature TGF-β1 dimer is non-covalently associated to the latency associated peptide (LAP). LAP forms a ring around mature TGF-β1, masking the interaction sites with the TGF-β1 receptor chains[14,15]. Only a few cell types are able to activate the cytokine by releasing mature TGF-β1 from LAP, through cell-type-specific mechanisms[16]. Upon T-cell receptor (TCR) stimulation, Tregs present latent TGF-β1 on their surface via disulfide linkage of LAP to a transmembrane protein called GARP[15,17,18]. Integrin αVβ8 interacts with GARP:(latent)TGF-β1 complexes, leading to release of active TGF-β1 close to the surface of stimulated Tregs[19,20]. Treg-derived active TGF-β1 exerts paracrine, short-distance immunosuppressive effects on immune cells, including T cells[16]. We derived anti-GARP:TGF-β1 mAbs that block the TGF-β1 activation by TCR-stimulated human Tregs, through a molecular mechanism elucidated via X-ray crystallography[15,21]. These mAbs do not bind complexes of GARP and latent TGF-β2 or β3, which are produced by non-Treg cells and once activated, signal via the same receptor as TGF-β1[15]. Blocking anti-GARP:TGF-β1 mAbs inhibited the immunosuppression by human Tregs of a xenogeneic graft-versus-host disease induced by transfer of human PBMCs into immunodeficient NSG mice[21].

Here, we derive an anti-mouse GARP:TGF-β1 mAb that blocks release of active TGF-β1 by mouse Tregs, allowing to examine the therapeutic benefit of blocking Treg function in tumor-bearing individuals. We show that this mAb increases the effector functions of anti-tumor T cells and induces immune-mediated rejections of tumors otherwise resistant to anti-PD-1 therapy. We also show that GARP-expressing Tregs are present in a sizeable subset of human melanoma samples, warranting trials to test anti-GARP:TGF-β1 mAbs in the clinics.

## Results

### Anti-GARP:TGF-β1 mAb blocks TGF-β1 activation by mouse Tregs.
Previously described blocking antibodies against human GARP:TGF-β1 complexes do not recognize mouse GARP:TGF-β1[21]. To derive mAbs that block TGF-β1 activation from GARP:TGF-β1 complexes on murine Tregs, we immunized llamas with plasmids encoding mouse GARP and TGF-β1, and constructed $V_H/V_k$ and $V_H/V_\lambda$ cDNA libraries to select Fab clones binding mouse GARP:TGF-β1 by phage display. Fab-encoding regions from selected clones were sequenced to construct >50 full-length mAbs by subcloning into a murine immunoglobulin G2a (mIgG2a) backbone. Clone 58A2 bound GARP:TGF-β1 complexes but not free GARP or free latent TGF-β1 (Fig. 1a). It bound the surface of mouse Tregs, both resting and even more so after TCR stimulation (Fig. 1b). It also blocked the release of active TGF-β1 induced by TCR stimulation of mouse Tregs in vitro (Fig. 1c), whether the mAb was used as a wild-type (WT) or an Fc-dead (FcD) mIgG2a subclass antibody. The FcD mIgG2a contains two amino-acid substitutions in the Fc region ($D_{265}A/N_{297}A$) that preclude binding to all mouse FcγRs[22,23] (Supplementary Fig. 1). Binding and blocking activities of clone 58A2 closely resembled those of blocking anti-human GARP:TGF-β1 mAbs[15,21]. Clone 58A2 was thus further selected to test whether treatment with blocking anti-GARP:TGF-β1 mAbs could improve anti-tumor immune responses and favor tumor rejections in mice.

### Anti-GARP:TGF-β1 overcomes resistance to anti-PD-1.
We injected CT26 colon carcinoma cells subcutaneously (s.c.) in BALB/c mice and started antibody treatments after 6 days, when tumors were well established in all mice (Fig. 2a). Tumors grew uniformly and no rejection (complete response or CR: 0/9) was observed upon injection of an isotype control mIgG2a antibody (Fig. 2b). No rejection was observed either after treatment with the blocking anti-GARP:TGF-β1 clone 58A2, either as a WT or an FcD mIgG2a. A single tumor rejection (CR: 1/10) was observed in mice treated with mAb 1D11, which neutralizes all three TGF-β isoforms[24]. Thus, antibodies that neutralize TGF-β1 or block TGF-β1 activation from GARP:TGF-β1 complexes do not display anti-tumor activity when administered as monotherapies in CT26 tumor-bearing mice (Fig. 2b).

Anti-PD-1 mAbs displayed very limited anti-tumor activity in this model (Fig. 2b). No rejection (CR: 0/10) occurred when anti-PD-1 clone RMP1-14 was administered as a rat IgG2a subclass mAb (WT), and only a minority of the mice (CR: 2/10) rejected their tumors after treatment with an anti-PD-1 comprising the RMP1-14 variable regions in an Fc-Silent (FcS) mIgG2a backbone (Absolute Antibodies®). Although minor, the increased anti-tumor activity of anti-PD-1 FcS compared to WT was expected because the FcS mAb contains amino-acid substitutions precluding binding to FcγRs, a feature known to enhance the anti-tumor activity of anti-PD-1 mAbs. In the case of clone RMP1-14, this has been suggested to result from abrogation of an FcγRIIb-dependent agonistic activity on PD-1 expressed by CD8+ T cells[25].

As shown in Fig. 2b, combination with anti-GARP:TGF-β1 WT improved the anti-tumor activity of anti-PD-1 in both WT and FcS formats (CR: 2/10 and 5/10 mice, respectively). Interestingly, the anti-tumor effect of anti-GARP:TGF-β1 did not require its binding to FcγRs: tumor rejection was also more frequent (CR: 4/10) when anti-GARP:TGF-β1 FcD was combined with anti-PD-1 FcS. Anti-TGF-β clone 1D11 modestly increased the frequency of tumor rejections (CR: 3/10) when combined with anti-PD-1 FcS. By comparison to treatment with anti-PD-1 FcS alone, reductions in mean tumor volumes were statistically significant in mice receiving anti-PD-1 FcS combined with anti-GARP:TGF-β1 (WT or FcD), but not with anti-TGF-β (Fig. 2c). This indicates that in CT26-bearing mice, blocking the activity of TGF-β1 emanating from GARP:TGF-β1-expressing cells only was at least as efficient as blocking the activity of the three TGF-β isoforms, whichever their cellular source. Anti-GARP:TGF-β1 significantly increased the anti-tumor activity of anti-PD-1 against established CT26 tumors in seven independent

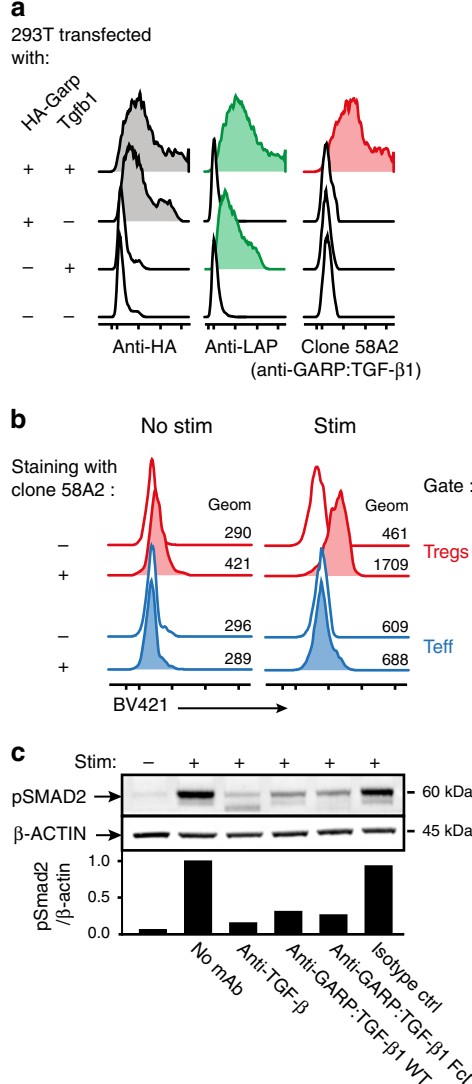

**Fig. 1 An anti-GARP:TGF-β1 mAb that blocks TGF-β1 activation by mouse Tregs in vitro. a** 293T cells were transfected with constructs encoding HA-tagged mouse GARP, mouse TGF- β1 (i.e. LAP + mature TGF- β1), or both, to induce surface expression of free GARP, free latent TGF- β1, or GARP:TGF- β1 complexes, respectively (Cuende et al.[21]). Transfected cells were stained with the indicated antibodies and analyzed by flow cytometry. Histograms are gated on live cells. **b** Mouse splenocytes were stimulated (stim) or not with anti-CD3/CD28 coated beads during 24 h, then analyzed by flow cytometry after staining with anti-CD4 and anti-FOXP3 antibodies in the presence (+) or absence (−) of the biotinylated anti-GARP:TGF-β1 clone 58A2, followed by streptavidin coupled to BV421. Histograms in red are gated on CD4+FOXP3+ cells (Tregs), in blue on CD4+Foxp3- cells (Teff). **c** Magnetically-sorted CD25+ mouse splenocytes were stimulated during 24 h with anti-CD3/CD28 coated beads in the presence or absence of blocking antibodies against active TGF- β (clone 1D11), GARP:TGF- β1 complexes (clone 58A2, mIgG2a WT or FcD), or an isotype control (mIgG2a WT), then analyzed by Western blot with antibodies against β-ACTIN and pSMAD2, as a read-out for active TGF-β1 production. Bar graphs on the bottom show quantification of ECL signals (ratio of pSMAD2/β-ACTIN signals relative to that in cells stimulated in the absence of blocking mAb). Full scans are shown in Supplementary Fig. 14.

experiments, allowing for a 2.8 to 5-fold increase of the proportion of mice surviving until the end of the experiment after having completely rejected their tumor (proportions of CR in meta-analyses shown in Fig. 3).

We verified whether our observations could be generalized to another tumor model, another genetic background, or yet other blocking anti-GARP:TGF-β1 mAbs. First, we injected MC38 colon carcinoma cells in WT C57BL/6 mice. We observed that anti-GARP:TGF-β1 increased the anti-tumor activity of anti-PD-1 in this model also (Supplementary Fig. 2). Second, we injected MC38 cells in *Garp*[YSG/YSG] C57BL/6 mice. These mice carry a homozygous knock-in mutation that replaces three contiguous HGN amino acids of mouse GARP by the YSG amino acids found at the corresponding positions of human GARP. Mutated GARP:TGF-β1 complexes can be bound by MHG-8 and LHG-10, two previously described blocking anti-human GARP:TGF-β1 antibodies[21]. Treatment of MC38-bearing *Garp*[YSG/YSG] mice with anti-PD-1 combined with MHG-8 or LHG-10 increased the frequency of tumor rejections by comparison to treatment with anti-PD-1 alone (Supplementary Fig. 3).

Altogether, these data indicate that antibody-mediated block-ade of TGF-β1 activation from GARP:TGF-β1 complexes induces rejection of tumors resistant to anti-PD-1 monotherapy. This anti-tumor activity does not require FcγR-dependent effector functions of the blocking anti-GARP:TGF-β1 mAbs.

**Anti-GARP:TGF-β1 protects against re-challenge with tumor.** Four mice that had rejected a CT26 tumor after treatment with a combination of anti-PD-1 WT and anti-GARP:TGF-β1 WT were injected 47 days after the last mAb administration with live CT26 cells in the right flank, and live RENCA or EMT6 syngeneic tumor cells in the left flank. No tumor grew in the right flanks whereas all grew in the left flanks (Fig. 4a). Control tumors grew readily in naive mice (Fig. 4b). This suggests that anti-GARP: TGF-β1 combined with anti-PD-1 induces protective T-cell-mediated immunity against CT26-specific tumor antigens.

**Anti-GARP:TGF-β1 does not modify numbers of TILs.** We examined whether treatment with anti-GARP:TGF-β1, alone or combined with anti-PD-1, modified the quantity or quality of the immune cells infiltrating CT26 tumors. Mice were injected sub-cutaneously with CT26 cells, treated with mAbs as indicated in Fig. 2a, and euthanized 1 day after the third mAb injection to collect spleens and tumors for flow cytometry, RNA analyses, or multiplexed immunofluorescence microscopy (Fig. 5, Supplementary Figs. 4–7). Tumor volumes and weights were measured the day before or immediately after dissection on day 13, respectively. On that early time point, tumor weights were already significantly reduced in mice treated with anti-GARP:TGF-β1 combined with anti-PD-1 by comparison to mice receiving the isotype control antibody (Fig. 5a and Supplementary Fig. 7a).

No difference in numbers and proportions of any leukocyte subset was observed in the spleens (Supplementary Fig. 4). Regardless of the treatment, all tumors contained ~9 × 10^4 leukocytes (CD45+ cells) per mm³ (Fig. 5b). Tumor-infiltrating leukocytes (TILs) comprised 16 ± 2% (mean ± sem) and 17 ± 2% of CD4+ and CD8+ T cells, respectively, and these proportions were not significantly different between the various treatment groups (Supplementary Fig. 5a). Likewise, numbers of these cells per mm³ of tumor (i.e. densities) were not significantly different between the various treatment groups (Fig. 5b). This was also observed in two similar independent experiments (Supplementary Fig. 6a). We also examined CD8+ T cells specific for a tumor

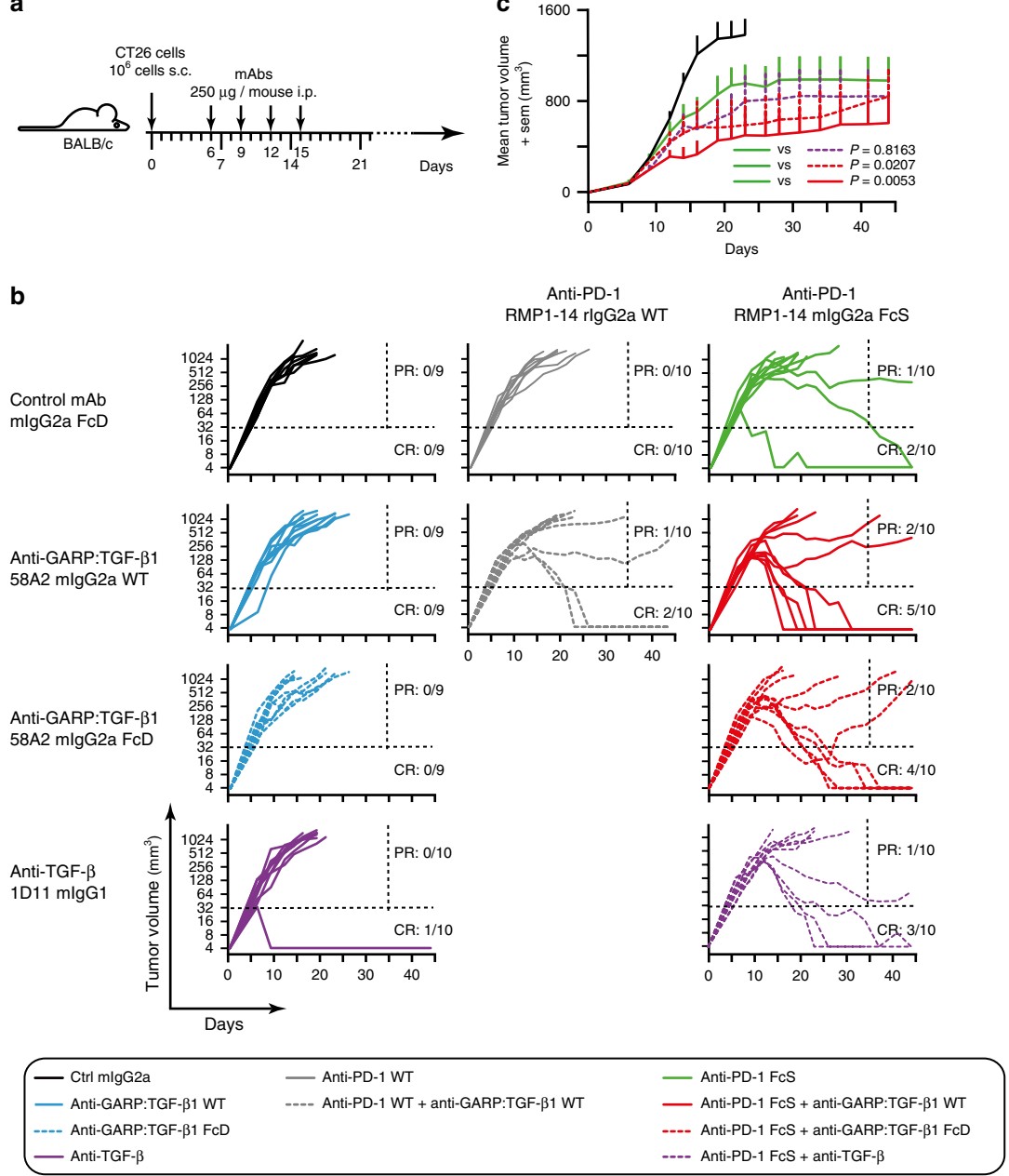

**Fig. 2 Combined blockade of GARP:TGF-β1 and PD-1 shows anti-tumor efficacy. a** Schematic representation of the experimental design. BALB/c mice were injected s.c. with live CT26 cells on day 0. Tumor diameters were measured twice a week. Mice were randomized in 10 groups ($n = 9$–10 mice per group) on day 6, and mAbs were injected i.p. every 3 days from day 6 to 15. Mice were euthanized when the tumor surface was ≥200 mm$^2$. **b** Evolution of tumor volumes in the 10 treatment groups (each line represents one mouse). Ratios indicate the proportions of complete responders (CR: mice alive on day 45 with no detectable tumor) and partial responders (PR: mice alive on day 35 that carry a tumor >32 mm$^3$ at the time of euthanasia). Dotted horizontal and vertical lines indicate the two arbitrary limits (tumor volume of 32 mm$^3$ and day 35) used to identify PR and CR. **c** Evolution of mean tumor volumes + sem in five treatment groups. *P* values for relevant comparisons are indicated and were calculated with a mixed effects model, as recommended for analyses of longitudinal data (Liu et al.[46]; Sugar et al.[45]), with a post-hoc Tukey's test for multiple comparisons. Similar results were observed in five other independent experiments (Fig. 3 shows meta-analyses of all pooled experiments).

antigen, using an AH1/H-2L$^d$ tetramer. AH1 is an immunodominant peptide presented by H-2L$^d$ on CT26 tumor cells. It is encoded by gene *Gp70*, an endogenous mouse retrovirus gene that is silent in normal mouse tissues and reactivated in CT26 cells[26]. Anti-AH1 CD8$^+$ T cells represented 31 ± 2% of the total CD8$^+$ T cells infiltrating tumors on day 13 (Supplementary Fig. 5a). Again, proportions and numbers of anti-AH1 CD8$^+$ T cells per mm$^3$ of tumor were not significantly different between

the various treatment groups (Fig. 5b and Supplementary Fig. 5a). Densities of total leukocytes or leukocyte subsets, including B, NK, or myeloid cells, were not modified in response to any treatment (Supplementary Fig. 6b).

We also estimated densities and distribution of CD4 and CD8 T cells in formalin-fixed paraffin-embedded (FFPE) tumor sections by immunofluorescence microscopy and quantitative digital imaging. Here again, we observed no significant difference

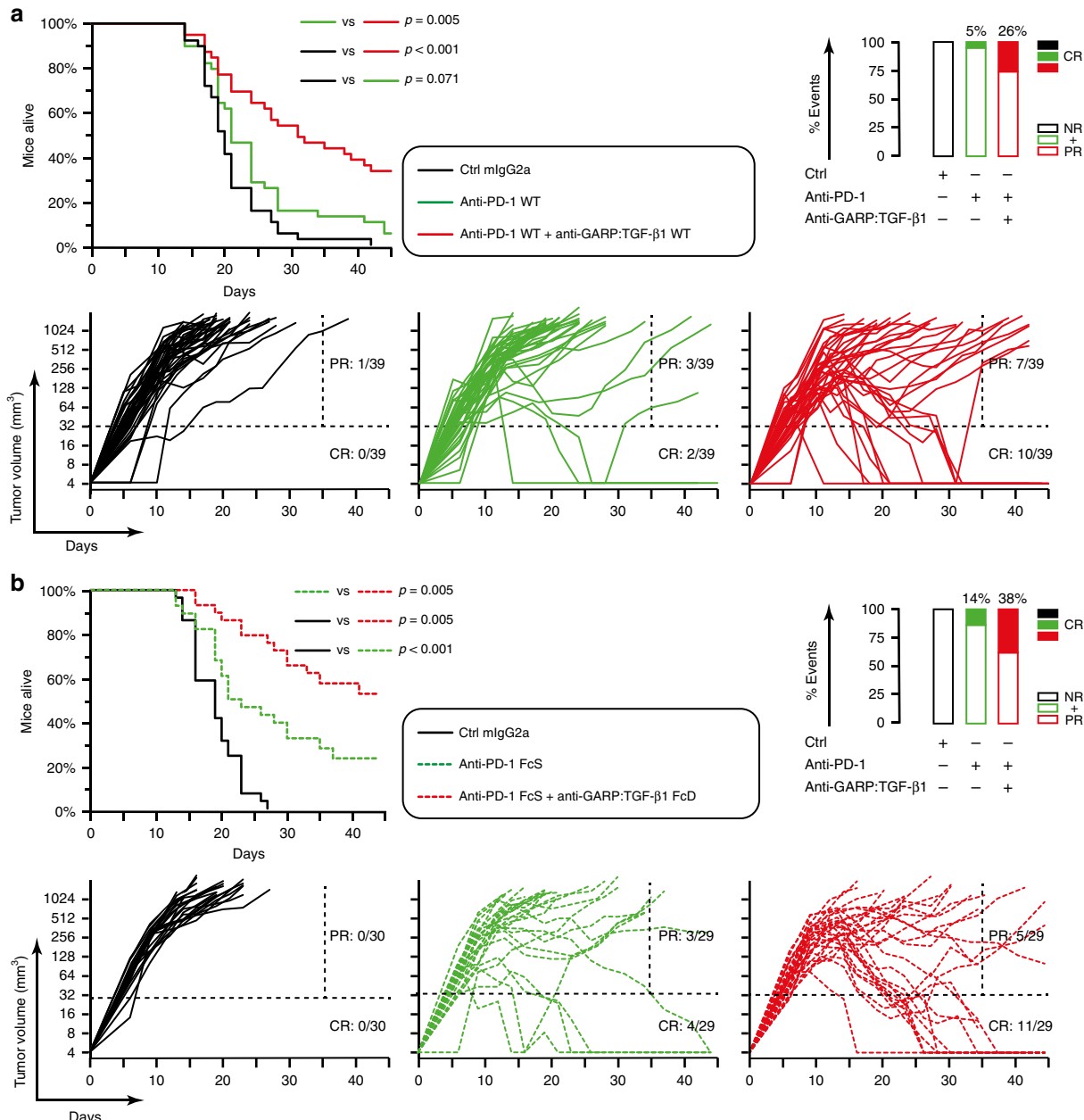

**Fig. 3 Combined blockade of GARP:TGF-β1 and PD-1 shows anti-tumor efficacy against established CT26 tumors.** BALB/c mice were injected s.c. with live CT26 cells on day 0. Tumor diameters were measured two to three times a week. On day 6, mice were randomized in various experimental groups and received the first of 3–4 mAb injections. Mice were euthanized when the tumor surface was ≥200 mm2. **a** Meta-analysis of four pooled independent experiments in which mice received anti-PD-1 WT alone, or in combination with anti-GARP:TGF-β1 WT (*n* = 39). **b** Meta-analysis of three pooled independent experiments in which mice received anti-PD-1 FcS alone, or in combination with anti-GARP:TGF-β1 FcD (*n* = 29–30). **a**, **b** Kaplan Meier plots on top represent the proportions of mice alive during the experiment, with *P* values calculated using a one-tailed Wilcoxon test. Graphs on bottom represent evolution of tumor volumes in various groups (each line represents one mouse). Ratios indicate the proportions of CR and PR, as defined in Fig. 2. Rectangles on the right represent proportions (%) of CR (filled rectangles), and partial + non-responders (PR + NR, empty rectangles).

between the various treatment groups (Supplementary Fig. 7a–c). A statistically non-significant trend toward increased densities of T cells, apparent in both the periphery and center of tumors, was observed in mice treated with anti-PD-1, whether it was combined or not with anti-GARP:TGF-β1 (Supplementary Fig. 7c).

Altogether, our results show that CT26 tumors are heavily infiltrated with leukocytes, including tumor-specific CD8+ T cells, which are nonetheless unable to control tumor growth in non-treated mice. Importantly, it also shows that the anti-tumor activity of the anti-GARP:TGF-β1 and anti-PD-1

combination does not result from an increased recruitment of anti-tumor T cells within these already inflamed tumors.

**Intra-tumoral Tregs are not depleted by anti-GARP:TGF-β1.** Flow cytometry analyses of cells isolated from CT26 tumors on day 13 indicated that 45 ± 3% of the CD4+ TILs were Tregs (FOXP3+), and 54 ± 2% of the tumor-infiltrating Tregs expressed GARP, regardless of the treatment (Supplementary Fig. 5a). We did not detect GARP on non-Treg TIL subsets, suggesting that the main target of anti-GARP:TGF-β1 mAbs within CT26 tumors

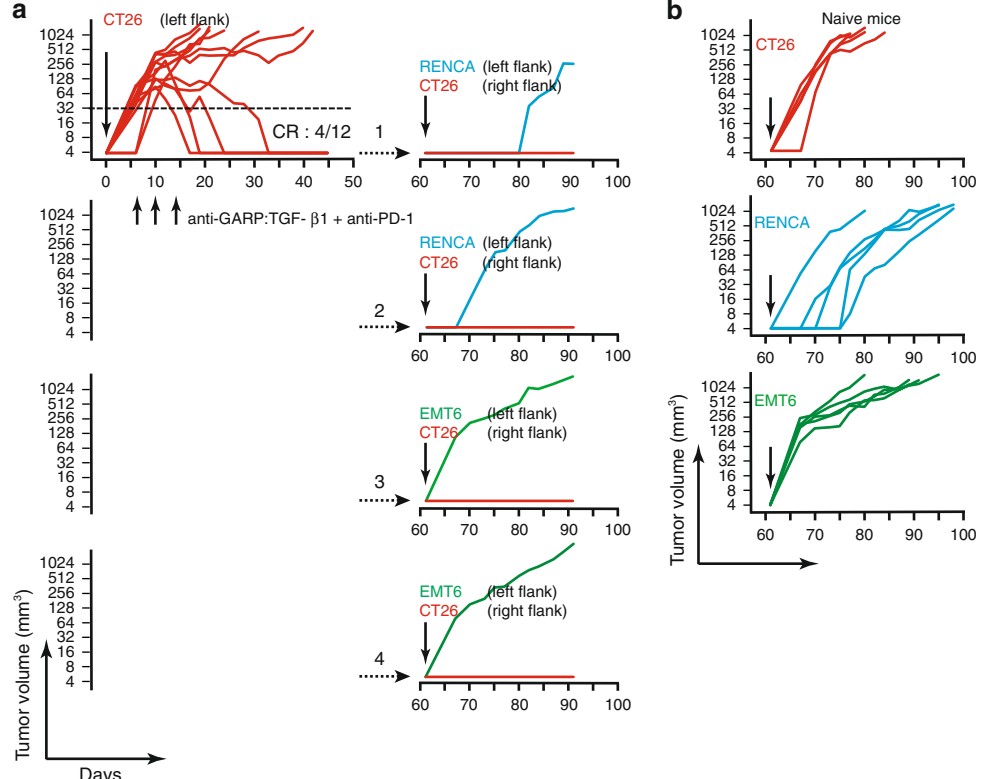

**Fig. 4 Combined blockade of GARP:TGF-β1 and PD-1 induces protective, CT26-specific immunity. a** On day 0, live CT26 cells were injected in the left flank of BALB/c mice (n = 12; 10^6 cells/mouse). Anti-GARP:TGF-β1 WT combined with anti-PD-1 WT were injected i.p. on days 6, 10, and 14. Graph on the left shows evolution of tumor volumes until day 50 in individual mice, and the ratio (CR) indicates the proportion of mice that completely rejected their tumor. On day 61, the four mice that had rejected their tumor were re-injected in the right flank with live CT26 cells (10^6 cells/mouse, red lines), and in the left flank with syngeneic RENCA (10^6 cells/mouse, blue lines) or EMT6 cells (3 × 10^5 cells/mouse, green lines). Each graph on the right shows evolution of tumor volumes in the left and right flanks in one of the four re-challenged mice. In all cases, mice were euthanized when the surface of any tumor was ≥200 mm². **b** On day 61, control naive BALB/c mice (n = 5 per group) were injected in the left flank with live CT26, RENCA or EMT6 tumor cells (cell numbers as in **a**). Graphs show evolution of tumor volumes in individual mice.

are Tregs. Nevertheless, numbers of total Tregs and GARP⁺ Tregs per mm³ of tumor were not decreased in mice that had received an anti-GARP:TGF-β1 mAb, alone or in combination with anti-PD-1 (Fig. 5c and Supplementary Fig. 6a). If anything, an increase in Treg and GARP⁺ Treg numbers was observed in mice treated with the anti-GARP:TGF-β1 FcD + anti-PD-1 combination in one experiment (Fig. 5c), but this was not confirmed in two others (Supplementary Fig. 6a). This indicates that anti-GARP:TGF-β1 antibodies, either in a WT or an FcD format, did not deplete intra-tumoral Tregs in this model.

**GARP:TGF-β1/PD-1 blockade augments anti-tumor T cell functions.** Anti-tumor CD8⁺ T lymphocytes promote anti-tumor immunity via production of pro-inflammatory cytokines such as TNFα and IFNγ and direct perforin/granzyme-dependent killing of tumor cells. Signaling induced in CD8⁺ T cells by binding of TGF-β1 to its receptor or PD-L1/L2 to PD-1 were both shown to inhibit these effector functions[27–29]. By comparison to mice injected with an isotype control mAb, significantly increased levels of *Prf1*, *Gzmb*, *Tnf*, and *Ifng* mRNAs were observed in tumors from mice treated with a combination of anti-PD-1 and anti-GARP:TGF-β1 WT or FcD mAbs, but not with either mAb alone (Fig. 5d). This suggested increased production of cytokines and cytolytic molecules by intra-tumoral T cells in response to the combination mAb treatment. We performed RNAseq and Gene Set Enrichment Analyses (GSEA[30–32]) to examine gene

expression signatures of response to pro-inflammatory cytokines within the tumor samples. By comparison to mice treated with the isotype control mAb, increased expression and significant enrichment of genes from the hallmark response signatures to IFNγ, TNFα, IL-2, and inflammation were observed in mice treated with the anti-GARP:TGF-β1 (WT or FcD) + anti-PD-1 combinations, but not with either monotherapy (Fig. 5e and Supplementary Fig. 8). This confirmed that combination therapy with anti-GARP:TGF-β1 and anti-PD-1 increased production of effector, pro-inflammatory cytokines within the tumors.

We next verified which immune cells produced more cytokines and cytolytic molecules in mice treated with the mAb combination. Cells isolated from CT26 tumors were left resting or briefly re-stimulated with peptide AH1 or PMA/Ionomycin in vitro, then stained for intracellular IFNγ and TNFα, and surface CD107a as a read out for degranulation and cytolytic activity. No significant difference in the proportions of cells expressing a single effector molecule, or a combination of these, was observed among total TILs, CD8⁺, or CD4⁺ T cells between the various treatment groups (Supplementary Fig. 9). In contrast, we observed significant increases in the proportions of cells expressing IFNγ, TNFα, or CD107a, and even more strikingly all three effector molecules, among the subset of CD8⁺ T cells directed against the tumor-specific AH1 antigen (AH1/H-2L^d tetramer positive cells) in tumors from mice treated with a combination of anti-PD-1 and anti-GARP:TGF-β1 WT or FcD mAbs, but not with either mAb alone (Fig. 5f and Supplementary

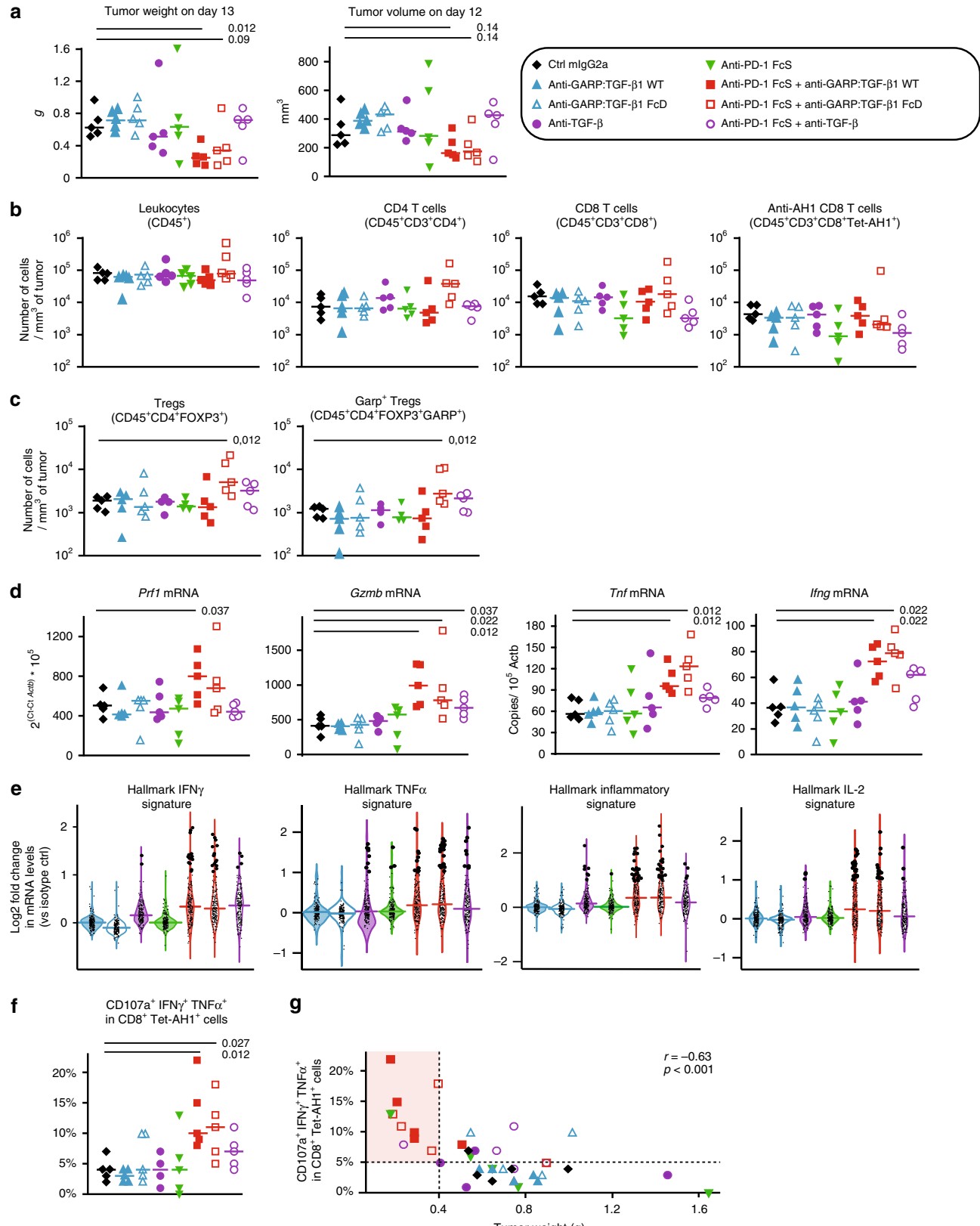

Fig. 9). This increase was observed whether or not cells had been re-stimulated in vitro with peptide AH1 or PMA/Ionomycin. Similar trends were observed when anti-PD-1 was combined with anti-TGF-β, although in that case, differences were in most cases not statistically significant (Fig. 5d–g and Supplementary Fig. 9).

Interestingly, tumor weights inversely correlated with proportions of anti-AH1 CD8[+] T cells with multiple effector functions (Fig. 5g). However, and as expected from above, tumor weights did not inversely correlate with densities of total leukocytes or any leukocyte subset (Supplementary Fig. 5b). Eight of the 10 mice

**Fig. 5 Combined blockade of GARP:TGF-β1 and PD-1 increases effector functions of anti-tumor CD8[+] T cells.** BALB/c mice ($n = 5$/group) were injected with CT26 cells on day 0 and treated with mAbs on days 6, 9, and 12, as illustrated in Fig. 2. Tumors were collected on day 13. **a** Weight of tumors collected after euthanasia on day 13 and volume of tumors measured the day before. Each data point represents the value measured in one mouse, and horizontal bars the median per group. **b**, **c** Numbers of various subsets of cells infiltrating 1 mm[3] of tumor, as determined by flow cytometry. Data points and horizontal bars as in **a**. **d** Expression of genes as determined by RT-qPCR. Data points and horizontal bars as in **a**. Numbers in italics show $P$ values < 0.05 for the comparisons with the control group (isotype ctrl mIgG2a), as calculated with a two-sided Wilcoxon test. **e** Violin plots representing fold change in expression of genes from hallmark signatures (MSigDB database) in each treatment group by comparison to the isotype control group, as determined by RNAseq. Within a group, each dot represents one gene of the indicated signature, with its position on the $Y$ axis representing the ratio between mean expression in the treated versus control group. Genes with a fold change ≥2 are represented by larger dots. Horizontal bars are median fold change for all genes of the signature. Violin contours show the kernel density ditributions of fold changes. Numbers of genes in the indicated signatures: IFNγ = 179; TNFα = 184; inflammatory = 164; and IL-2 = 175. **f** Cells isolated from tumors were stimulated in vitro with the AH1 peptide, and analyzed by flow cytometry for surface markers and intracellular cytokines. Data points, horizontal bars, and $P$ values as in **d**. **g** Correlation between proportions of anti-tumor CD8[+] T cells displaying multiple effectors functions (shown in **f**) with tumor weight in the corresponding mouse at day 13. ϱ = Pearson's correlation coefficient with corresponding $P$ value calculated with one-tailed $F$-test. Results shown here are representative of at least three independent experiments.

with the smallest tumors (<0.4 g) had ≥5% of anti-AH1 CD8[+] T cells with multiple effector functions and received the combination of anti-PD-1 and anti-GARP:TGF-β1 WT or FcD mAbs (Fig. 5g).

Altogether, these data indicate that treatment with a combination of anti-GARP:TGF-β1 and anti-PD-1 increases the effector functions of tumor-specific CD8[+] TILs without increasing their number or penetration within the tumors.

**Anti-tumor activity requires CD8+ cells and IFNγ signals.** We next sought to determine whether increased effector functions of anti-tumor CD8[+] TILs contributed to the anti-tumor activity of the anti-GARP:TGF-β1 and anti-PD-1 combination. First, we administered a monoclonal antibody that depletes CD8[+] cells (Supplementary Fig. 10) into tumor-bearing mice, 2 days before starting the treatment with anti-GARP:TGF-β1 WT and anti-PD-1 WT mAbs. Depletion of CD8[+] T cells abrogated the anti-tumor efficacy of the combination treatment (Fig. 6a). Second, we administered a neutralizing anti-IFNγ mAb into tumor-bearing mice on the same days as treatments with anti-GARP:TGF-β1 and anti-PD-1. Neutralization of IFNγ also abrogated the anti-tumor efficacy of the combination treatment (Fig. 6b). Together, these results suggest that increased IFNγ production by CD8[+] T cells contributes to the anti-tumor activity of anti-GARP:TGF-β1 combined with anti-PD-1.

**Anti-GARP:TGF-β1 acts by blocking TGF-β1 activation on Tregs.** In CT26, GARP expression is detected on a majority of Tregs (Fig. 5) but not on other TIL subsets. This suggests that when combined with anti-PD-1, anti-GARP:TGF-β1 exerts anti-tumor activity by blocking TGF-β1 activation by Tregs but not by other GARP-expressing cell types. However, TGF-β1 activation by GARP[+] platelets was also suggested to suppress anti-tumor immunity[33]. We thus derived two C57BL/6 mouse strains carrying a Treg- or a platelet- specific deletion of the *Garp* gene, respectively (Supplementary Fig. 11). We injected MC38 cells in Treg- and platelet-specific *Garp* KO mice and their WT littermates, and treated tumor-bearing mice with anti-PD-1 combined or not with anti-GARP:TGF-β1 (Fig. 7a). As shown in Fig. 7b, the proportions of complete responses to anti-GARP:TGF-β1 + anti-PD-1 were superior to anti-PD-1 alone in platelet-specific *Garp* KO mice, as well as in their WT littermates (42% vs 23%, and 46% vs 13%, respectively). These results were in line with our previous experiments but did not reach statistical significance. In Treg-specific *Garp* KO mice, this difference was not observed (Fig. 7c), and if anything, the combination was modestly inferior to anti-PD-1 alone (not statistically significant). These results indicate that targeting GARP on Tregs, but not platelets, with a

blocking anti-GARP:TGF-β1 mAb is necessary to overcome resistance to PD-1 blockade in tumor-bearing mice. Our observation that the anti-tumor activity of anti-PD-1 alone is only very modestly increased in Treg-specific *Garp* KO mice by comparison to WT littermates (CR: 28% vs 20%) suggests that GARP-deficient Tregs may acquire compensatory immunosuppressive mechanisms during differentiation that inhibit immune responses against experimentally transplanted tumors.

**GARP+ Tregs are present in human melanoma metastases.** The results above suggest that anti-GARP:TGF-β1 mAbs could exert anti-tumor effects and increase response rates to PD-1/PD-L1 blockade, particularly in patients with tumors containing GARP-expressing Tregs. We thus resorted to multiplex immunofluorescence (mIF) staining to assess the presence and abundance of GARP[+]FOXP3[+] cells in a serie of 19 melanoma samples (Fig. 8a–c). Double positive cells correspond to activated Tregs in human tissues: even though FOXP3 can be expressed in non-Treg T cells, GARP expression is induced by TCR stimulation in Tregs, but not in other T cells[18]. We tested many anti-GARP mAbs but we could not detect specific GARP staining on sections from FFPE tissue samples. In contrast, we observed specific staining on sections of frozen cells or tissues, using three different anti-GARP mAbs (clones Plato-1, MHG-6 and LHG-10; Supplementary Fig. 12). In frozen tonsil tissue, GARP immunoreactivity was observed in a subset of FOXP3[+] cells, but also mainly in the blood vessel wall including the endothelium, as indicated by dual CD34 staining. Tumors displayed a similar staining pattern, including GARP[+] blood vessels and GARP[+]FOXP3[+] Tregs. FOXP3[+] cells (further referred to as Tregs, even though these cells may also include some activated, non-Treg T cells) and GARP[+]FOXP3[+] Tregs were enumerated by quantitative digital imaging (Fig. 8b, c). Intra-tumoral Treg abundance varied greatly, ranging from almost zero per 10[5] nuclei, to similar counts to tonsil tissue (±5000 per 10[5] nuclei). Altogether, a third (6/19) of cutaneous melanoma metastases contained ≥1000 Tregs and ≥50 GARP-expressing Tregs per 10[5] nuclei (Fig. 8c). RNAseq analysis was performed on frozen fragments derived from the same 19 metastasis samples (Fig. 8a). Proportions of Tregs and GARP[+] Tregs as determined by mIF correlated strongly with *FOXP3* gene expression (Fig. 8d), thereby validating the observations from staining and counting. They also correlated strongly with T-cell genes such as *CD3G* (Fig. 8d), and IFNγ responsive genes, indicating that the more T cells and activated IFNγ-producing T cells, the more Tregs and GARP[+] Tregs in the melanoma samples (Supplementary Data 1). We used GSEA to determine whether specific gene signatures were enriched in genes whose expression correlated most closely with proportions of GARP[+] Tregs in the melanoma samples. The gene signatures were obtained either

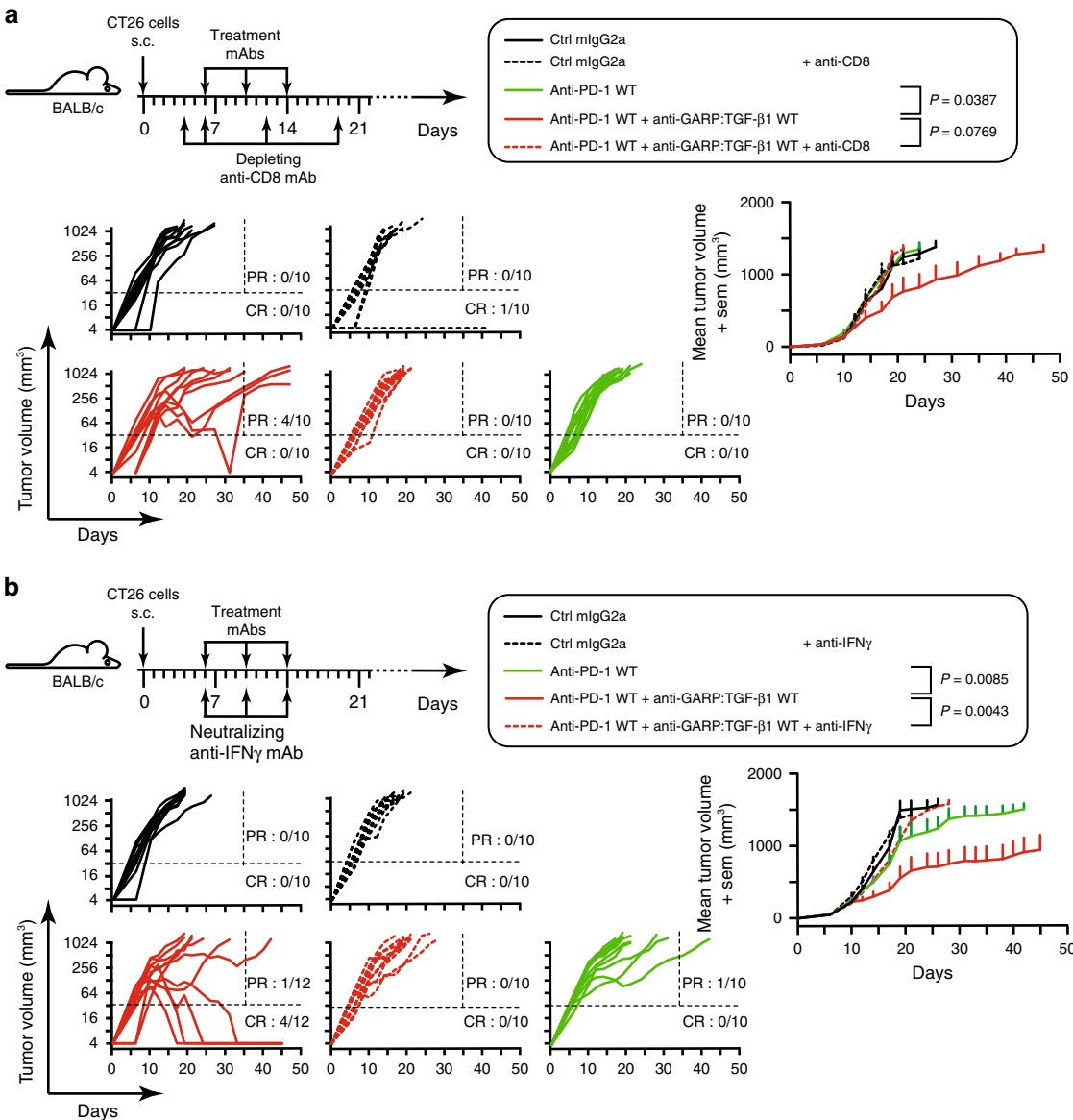

**Fig. 6 Anti-tumor efficacy of combined blockade of GARP:TGF-β1 and PD-1 requires CD8+ cells and IFNγ. a** Experimental design as in Fig. 2, except that mice (n = 10–12/group) also received injections of a depleting anti-CD8 antibody at the indicated time points. Graphs on the left show evolution of tumor volumes in individual mice. Ratios indicate the proportions of CR, (mice alive at the end of the experiment with no detectable tumor) and PR (mice alive on day 35 that carry a tumor >32 mm³ at the time of euthanasia). Dotted horizontal and vertical lines indicate the two arbitrary limits (tumor volume of 32 mm³ and day 35) used to identify PR and CR. Graph on the right shows evolution of mean tumor volumes + sem in the five treatment groups. P values for relevant comparisons are indicated in the graphical legend and were calculated with a mixed effects model (Liu et al.[46]; Sugar et al.[45]), with a post-hoc Tukey's test for multiple comparisons. **b** As in **a**, except that mice received a neutralizing anti-IFNγ.

from the MSigDB public database (hallmark gene sets, http://software.broadinstitute.org/gsea/msigdb/) or from our own experimental data (experimental gene sets, Supplementary Data 2). Hallmark signatures of response to inflammation, IFNγ, TNFα, or IL-2 were significantly enriched at the left end of the gene list ordered by correlation with proportions of GARP+ Tregs (Enrichment Score >0.5 and False Discovery Rate <0.1%; Fig. 8e). Similar enrichments were observed for experimental gene sets of IL-1β-, TNFα-, or TGF-β1-response induced in human melanoma cell lines, primary endothelial cells, fibroblasts, melanocytes, or CD4+ T cell clones (Fig. 8f). Altogether, this indicates that GARP+ Tregs were mostly found in inflamed melanoma metastases infiltrated by activated T cells, and were associated with higher levels of TGF-β signaling, including TGF-β signaling within the T-cell compartment.

## Discussion

Our observations in tumor-bearing mice and human cutaneous melanoma metastases suggest that anti-GARP:TGF-β1 mAbs could serve as an immunotherapeutic approach to inhibit TGF-β1-dependent immunosuppression by intra-tumoral Tregs in patients with cancer.

In mice bearing CT26 or MC38 tumors, blocking anti-GARP:TGF-β1 mAbs did not induce tumor regression when administered alone, in line with observations in mice carrying a Treg-specific deletion of the *Garp* gene, which did not show reduced growth of MC38 or GL261 tumors by comparison to WT (Fig. 7 and Vermeersch et al.[34]). But when combined with anti-PD-1 mAbs, anti-GARP:TGF-β1 mAbs significantly increased the frequency of tumor rejection relative to anti-PD-1 alone. This does not necessarily imply that anti-GARP:TGF-β1 mAbs will have no

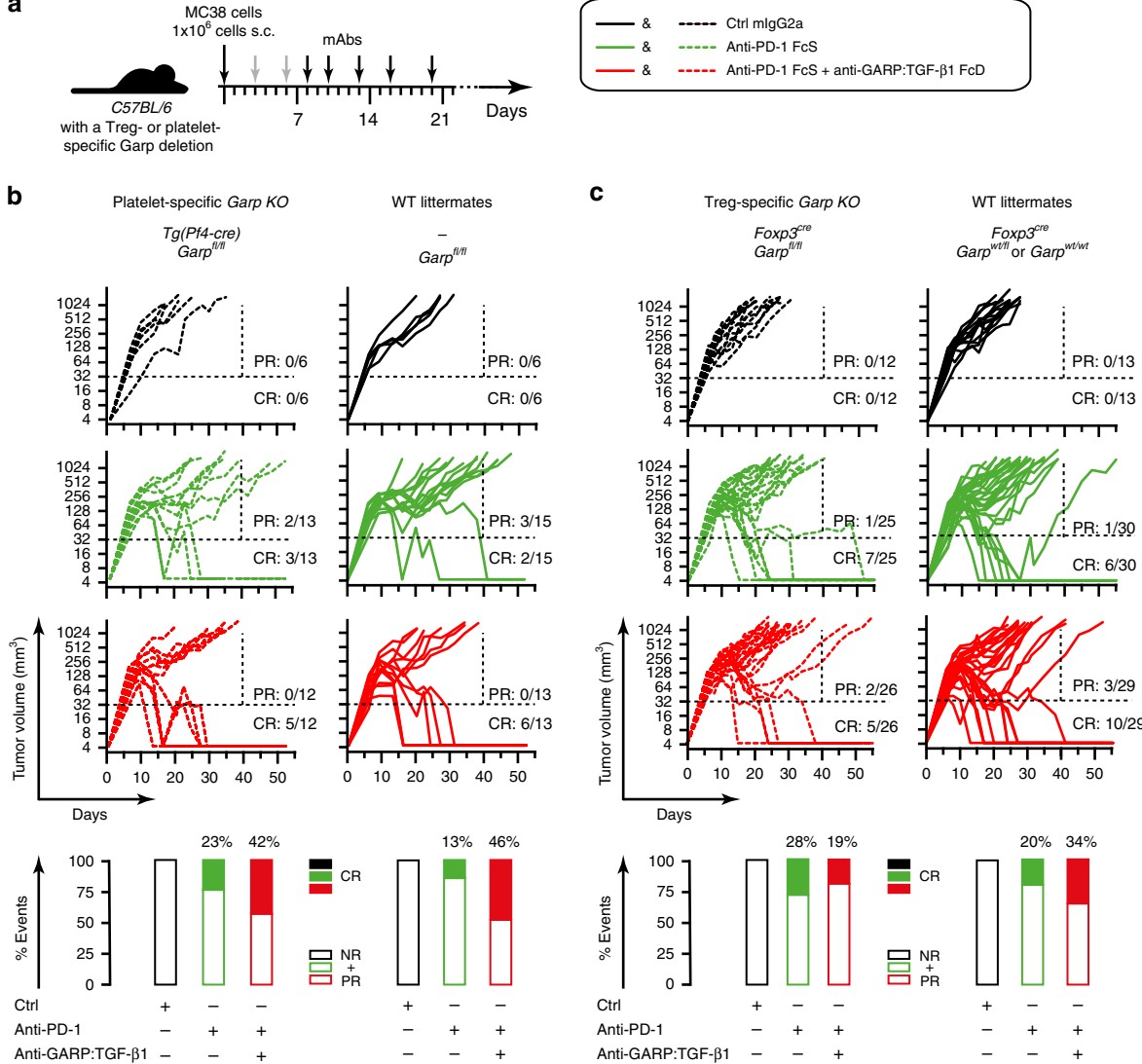

**Fig. 7 Targeting GARP on Tregs is sufficient to increase the anti-tumor activity of anti-PD-1 blockade. a** Schematic representation of the experimental design. Genetically modified C57BL/6 mice were injected s.c. with live MC38 cells on day 0. Anti-GARP:TGF-β1 and isotype control mAbs (250 μg/mouse) were injected i.p. on day 3, 6, 8, 10, 13, 16, and 20 (gray and black arrows). Anti-PD-1 mAb (25 μg/mouse) was injected on day 8, 10, 13, 16, and 20 (black arrows). Mice were euthanized when the tumor surface was ≥200 mm². **b** Evolution of individual tumor volumes in groups of mice carrying a platelet-specific deletion of *GARP* (left) and their littermate controls (right). **c** Same as **b**, for mice carrying a Treg-specific deletion of *GARP* (left) and their littermate controls (right). Results pooled from two independent experiments. **b, c** Each line represents one mouse. Ratios indicate the proportions of CR (live mice with no detectable tumor at the end of the experiment) and PR (mice alive on day 40 that carry a tumor >32 mm³ at the time of euthanasia). Dotted horizontal and vertical lines indicate the two arbitrary limits (tumor volume of 32 mm³ and day 40) used to identify PR and CR. Rectangles on the bottom represent proportions (%) of CR (filled rectangles), and partial + non-responders (PR + NR, empty rectangles).

anti-tumor activity as monotherapy in patients with cancer, but it does suggest that they are able to overcome primary or acquired resistance to PD-1/PD-L1 blockade.

Our experiments in cell-specific *Garp* KO mice suggest that blocking the activity of TGF-β1 emanating from GARP-expressing Tregs is required for anti-GARP:TGF-β1 to exert anti-tumor activity. They suggest also that blocking the activity of TGF-β1 emanating from GARP-expressing platelets or endothelial cells is neither necessary nor sufficient, although this requires further investigation.

In WT mice, combination therapy required CD8+ cells and IFNγ signals for efficacy, and increased the expression of multiple effector molecules by anti-tumor CD8+ TILs. Notably, tumors from untreated mice were already heavily infiltrated by immune cells, and densities of TILs or TIL subsets (including anti-tumor

CD8+ T cells) were not modified by anti-GARP:TGF-β1, anti-PD-1, or a combination of the two. This suggests that in MC38 or CT26 models at least, blocking the activity of TGF-β1 emanating from GARP-expressing Tregs triggers tumor regression by inducing or re-invigorating inflammatory and cytolytic activities of anti-tumor CD8+ T cells that are already present in the tumor.

This mode of action strikingly differs from those reported by Mariathasan et al.[35] and Dodagatta-Marri et al.[36] for neutralizing anti-TGF-β mAbs combined with PD-1/PD-L1 blockade. In the former report, therapeutic co-administration of anti-TGF-β and anti-PD-L1 mAbs was found to reduce TGF-β signaling in stromal cells such as fibroblasts and increase penetration of CD8+ T cells into EMT6 tumors, thus provoking anti-tumor immunity and tumor regression[35]. In the latter report, anti-TGF-β combined with anti-PD-1 was suggested to act by blocking TGF-β

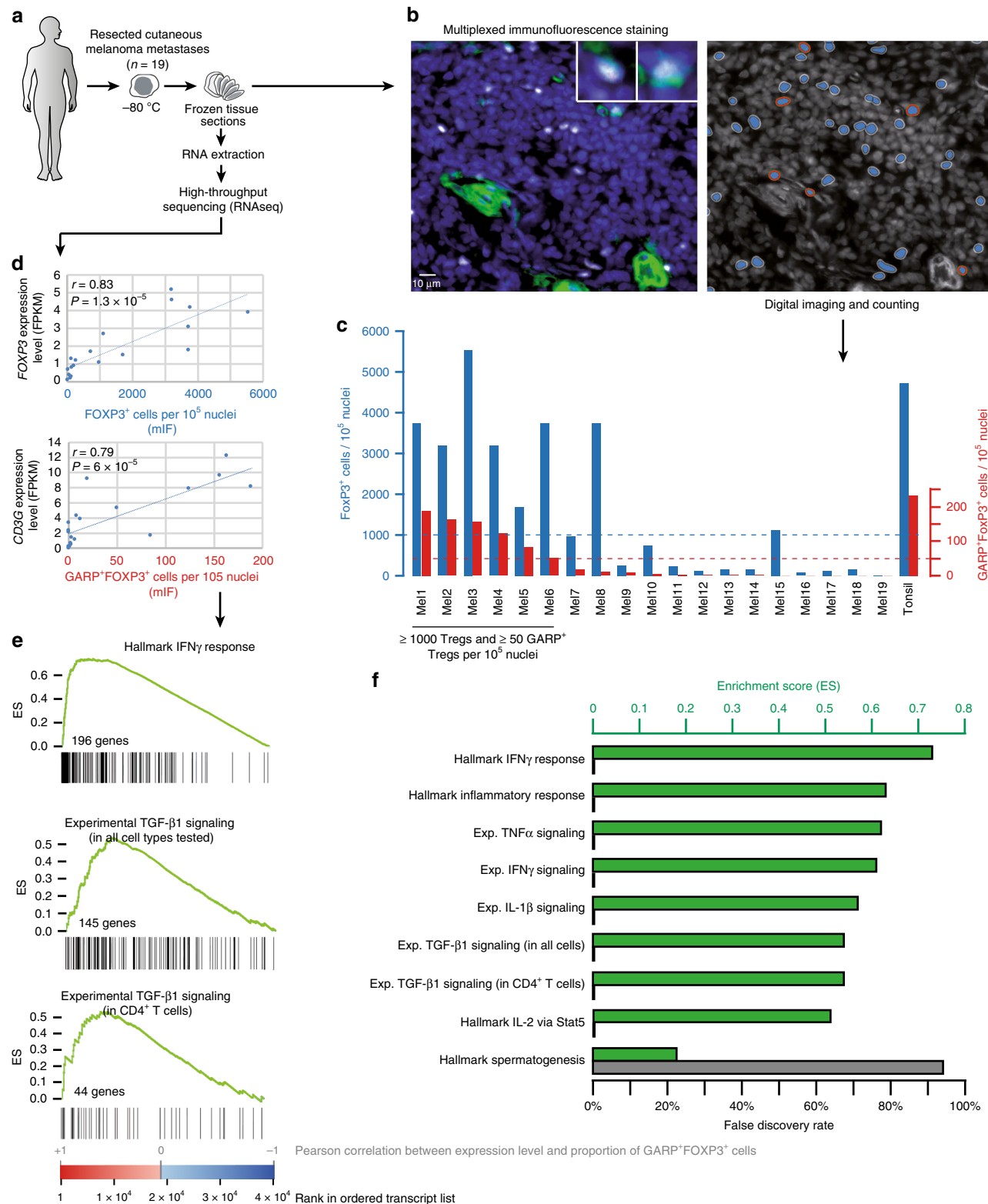

activity on CCK168 tumor cells and intra-tumoral Tregs. This was based on observations that anti-TGF-β reduced the tumor cell SMAD3 phosphorylation and Treg/Th ratios, which were increased by treatment with anti-PD-1 alone[36]. Variation in the proposed modes of action of anti-GARP:TGF-β1 and anti-TGF-β may result from different tumor models being used in our laboratories. It could result also from the different sources of TGF-β activity blocked by the various mAbs: anti-GARP:TGF-β1

only blocks TGF-β1 emanating from GARP-expressing cells such as Tregs, whereas anti-TGF-β neutralizes activity of all three TGF-β1, β2, and β3 isoforms, regardless of their cellular source. Our data suggest that blocking only the TGF-β1 produced on Treg surfaces with anti-GARP:TGF- β1 mAbs is sufficient to increase the anti-tumor activity of PD-1/PD-L1 targeting, while exerting less undesired effects on the tumor microenvironment than that induced by broad blockade of all TGF-β activity. In

**Fig. 8 Association between inflammatory gene signatures and activated Tregs in a series of 19 human melanoma metastases. a** Experimental setting. See text for details. **b** FOXP3 and GARP-expressing cells were stained by mIF on frozen tumor sections. Left panel: representative view from tumor sample Mel1. Nuclei appear in blue, FOXP3 in white and GARP in green. Upper right: detail of two FOXP3[+]GARP[+] cells. Right panel: automated detection of FOXP3[+]GARP[−] (blue) and FOXP3[+]GARP[+] cells (blue ringed red). **c** Diagram representing the proportion of FOXP3[+] and FOXP3[+]GARP[+] cells in each tumor sample, calculated from the automated counts of nuclei, FOXP3[+] and FOXP3[+]GARP[+] cells. **d** Top: graph showing the Pearson correlation between the proportion of FOXP3[+] cells and the level of expression of the *FOXP3* gene, obtained by RNAseq analysis of tissue sections from the same tumors. Bottom: idem with FOXP3[+]GARP[+] cells and *CD3G*. Indicated *P*-values for Pearson correlation were calculated with a two-sided *t*-test **e** Gene set enrichment analysis (GSEA). The 38,602 transcripts measured by RNAseq were ordered by decreasing Pearson correlation between their expression level and the proportion of FOXP3[+]GARP[+] cells in the 19 tumor samples. Black vertical bars indicate positions of genes from various gene sets in the ordered transcript list. Gene set enrichment in the ordered transcript lists are plotted as green curves, with the Enrichment Score (ES) corresponding to the maximum value. Upper panel: gene set of the hallmark IFNγ signature (MSigDB). Middle and bottom panels: gene sets induced by TGF-β1 as determined in expression microarray experiments in which human melanoma cell lines, primary endothelial cells, fibroblasts, melanocytes, or a CD4[+] T cell clone were exposed to the recombinant cytokine (middle panel: genes induced by TGF-β1 in at least one of the five cells types; bottom panel: genes induced by TGF-β1 in CD4[+] T cells; See also Supplementary Data 2). **f** ES (green rectangles) and false discovery rate (gray rectangles) measured by GSEA for the indicated gene sets. The false discovery rate was obtained by calculating ES for 1000 gene-set permutations.

human melanoma samples infiltrated by T cells and GARP-expressing Tregs, we found evidence of TGF-β signaling within the T cell compartment, suggesting that blockade of Treg-derived TGF-β1 activity with anti-GARP:TGF-β1 mAbs may be sufficient to increase CD8[+] T-cell-mediated anti-tumor immunity, while avoiding potential toxicity associated with a more global inhibition of TGF-β signaling.

FcγR-dependent functions of immunostimulatory mAbs affect their anti-tumor activity in different ways. Anti-CTLA-4 and anti-PD-L1 mAbs must bind activating FcγRs to exert potent anti-tumor activity by ADCC- or ADPC-mediated depletion of Tregs, or myeloid and tumor cells, respectively[10,25,37–39]. On the contrary, anti-PD-1 mAbs are more potent in formats that do not bind FcγRs, because this prevents cross-linking and agonistic activity or depletion of PD-1-expressing anti-tumor CD8[+] T cells[25,40]. Here, we establish that anti-GARP:TGF-β1 mAbs do not require FcγR-dependent functions to exert anti-tumor activity in mice, supporting a mode of action by which they block TGF-β1 activation and downstream signaling without depleting GARP-expressing cells. This suggests that in cancer patients, blocking anti-GARP:TGF-β1 mAbs that are unable to bind FcγRs would be as efficient and probably safer than effector-competent formats, because they would not cause Treg depletion and potential subsequent auto-immune adverse events, nor kill any other GARP-expressing cells in cutaneous melanoma metastases and non-cancerous tissues.

Taken together, our results support the clinical evaluation of blocking anti-GARP:TGF-β1 mAbs, administered alone or in combination with other therapeutic strategies, to treat patients with cancer resistant to currently available immunotherapies. A phase I trial was recently initiated to test such antibodies in the clinics (ClinicalTrials.gov: NCT03821935).

## Methods
**Mice.** BALB/c, *Garp^YSG/YSG* C57BL/6, Treg-specific-, and platelet-specific-*Garp* KO mice were bred at the SPF animal facility of the UCLouvain. Cell type specific-*Garp* KO and WT littermates were obtained by crossing *Lrrc32^tm1.1Hfuj* with *B6.129 (Cg)-Foxp3^tm4(YFP/icre)Ayr/J* or *Tg(Pf4-icre)Q3Rsko* mice. WT C57BL/6J mice used in Fig. S3 mice were bred at the conventional animal facility of the KULeuven. The facility is controlled to maintain temperature between 20 and 24 °C; HR between 40 and 65% and day–night cycles of 12h–12h. All animal studies were performed in accordance with national and institutional guidelines for animal care, under permit numbers 2015/UCL/MD/19 and 2019/UCL/MD/032 at the UCLouvain, and 263–2014 at the KULeuven.

**Cell lines.** CT26 colon carcinoma, EMT6 breast carcinoma, RENCA renal carcinoma, MC38 colon adenocarcinoma and 293T cells were maintained in vitro as a monolayer culture in Iscove's Modified Dulbecco Medium (CT26), RPMI (EMT6 and RENCA), or Dulbecco's Modified Eagle Medium (293T and MC38), supplemented with 10% fetal calf serum (FCS) and 10 mM Hepes, 1 mM sodium

pyruvate, and 0.1 mM non-essential amino acids, at 37 °C in an atmosphere of 8% or 5% $CO_2$ in air. Murine tumor cells in exponential growth phase were harvested, washed in PBS, and resuspended in endotoxin-free Dulbecco's PBS (Millipore) prior to s.c. inoculation into mice.

**Anti-GARP:TGF-β1 mAb 58A2.** To determine whether clone 58A2 bound free GARP, GARP:TGF-β1 complexes or both, 293T cells were transiently transfected with plasmids encoding HA-tagged mouse GARP, mouse TGF-β1, or both, and analyzed by flow cytometry 24 h after transfection with anti-HA, anti-LAP, or mAb 58A2. Binding of 58A2 to murine Tregs was evaluated by performing flow cytometry on mouse splenocytes that had been stimulated in vitro or not during 24 h with anti-CD3/28 coated beads (ThermoFisher). Antibodies used for staining were: anti-CD4, anti-FOXP3, biotinylated anti-GARP:TGF-β1 58A2, and streptavidin coupled to BV421. The ability of 58A2 to block active TGF-β1 production by murine Tregs was evaluated using CD25[+] cells isolated from mouse spleens with anti-CD25 microbeads and an AUTOMacs sorter (Miltenyi) as a source of murine Tregs. Sorted CD25[+] cells were highly enriched in Tregs and contained ±70% of CD4[+]FOXP3[+] cells as determined by intracellular FACS analyses. Sorted CD25[+] cells were stimulated in vitro with anti-CD3/CD28 coated beads during 24 h in presence or absence of 20 µg/ml of clone 58A2 (mIgG2a WT or FcD), or control antibodies (mIgG2a isotype control Motavisumab, or anti-TGF-β), and were analyzed by Western blot with antibodies against β-ACTIN (Sigma) or phosphorylated SMAD2 (Cell Signaling Technology) as a read-out for active TGF-β1 production[21]. ECL signals were quantified using the Bio1D software. Full scans of Western blots are shown in Supplementary Fig. 14.

**Binding of mAbs to Fcγ receptors.** Recombinant biotinylated extracellular domains of immunoglobulin Fcγ receptors (FcγRs) were purchased from Sino-Biologicals. Interactions of mAbs with FcγRs were measured using the Biacore T200 (GE Healthcare). Briefly, the biotin CAPture kit was used to immobilize biotinylated FcγRs to the sensor chip. Prior to measurements, one conditioning cycle consisting of three serial regeneration steps (1 min, flow rate 10 µL/min) was performed. In each measurement cycle, Biotin CAPture reagent was applied to the reference and measurement channel (5 min, flow rate 2 µL/min). Biotinylated FcγRs at a concentration of 1 µg/mL were injected (2 min, flow rate 10 µL/min) in the active flow cell only. For each kinetics experiment, five dilutions of purified mAb were applied at concentrations in the range of 6.2–500 nM for binding to FcγRI, 12.8–8000 nM for FcγRII/III, and 0.8–8000 nM for FcγRIV. Interaction analysis was performed in the single-cycle kinetic mode (2 min at 30 µL/min) followed by 10 min of dissociation. Flow cells were regenerated (2 min, 20 µL/min) with a 6-M guanidine-HCl/0.25 M NaOH solution. Data were collected using dual detection at 10 Hz and analyzed using the Biacore T200 Evaluation Software.

**Animal experiments.** On day 0, live CT26 cells ($10^6$ cells/mouse) or MC38 cells ($1.5$ or $0.5 \times 10^6$ cells/mouse in Supplementary Figs. 3 and 4, respectively) were injected s.c. into 6- to 12-week-old syngeneic mice. Large (D) and small (d) tumor diameters were measured with a caliper every 2 or 3 days starting on day 6. Mice were euthanized for ethical reasons when the tumor surface (D × d) reached 200 mm² (Figs. 2, 3, 4, 6, 7 and Supplementary Fig. 4), or when the tumor volume reached 1300 mm³ or the surface was ulcerated (Supplementary Fig. 3). Tumor volumes were calculated as follows: $V = \pi \times D \times d^2/6$. When tumors were not palpable or too small to be measured (i.e. prior to day 6 or after complete tumor rejection), volumes were arbitrarily set to 4 mm³. On days indicated in the figure legends, mice received intraperitoneal (i.p.) injections of the following mAbs, administered alone or combined as indicated in the figure legends: isotype control (motavizumab), anti-GARP:TGF-β1 (clone 58A2, mIgG2a WT or FcD), anti-PD-1 (WT or FcSilent), or anti-TGF-β. Once a tumor reaches the maximum tolerated size (pre-defined as an ethical humane endpoint in our protocol), the

corresponding mouse was euthanized. The last tumor size measured prior to euthanasia (i.e. maximum size) is carried forward in the data series for later time points. Average tumor size per group is calculated for all tumors, including the largest tumors, at all time points. As soon as all mice in a given group have been euthanized, the average size is not computed anymore for that group, and the average growth curve is interrupted on the graphs.

For re-challenge experiments, BALB/c mice that had rejected CT26 tumors injected in their left flank on day 0 were re-injected s.c. with live tumor cells in both flanks on day 61. CT26 cells ($10^6$ cells/mouse) were re-injected in the right flank, and RENCA cells ($3 \times 10^5$ cells/mouse) or EMT6 cells ($10^6$ cells/mouse) were injected in the left flank.

In some experiments, mice received i.p. injections of a depleting anti-CD8 mAb (0.5 or 0.25 mg/mouse, as indicated in the legend of Fig. 6) on days 4, 6, 12, and 19 after tumor cell inoculation. In other experiments, mice received i.p. injections of a neutralizing anti-IFNγ mAb (clone FX4F3[41], 0.25 or 0.1 mg/mouse) on days 6, 10, and 14.

**Monitoring of immune responses in tumor-bearing mice.** On day 13 after inoculation of tumor cells performed as indicated above, tumors were harvested and mechanically dissociated in the presence of enzymes (Collagenase I 100 mg/ml, Life Tech; Collagenase II 100 mg/ml, Life Tech; Dispase 1 mg/ml, Life Tech; and DNAse I 0.4U/ml, Roche), using two cycles in the GentleMacs disruptor (Miltenyi) separated by 30 min of incubation at 37 °C. Tumor cell homogenates were clarified through 70 μm and 40 μm filters. Single-cell suspensions were counted on a Luna® cell counter with a live-dead cell marker, then pelleted, and resuspended either in PBS containing 2 mM EDTA and 1% FCS for immediate staining, or in X-Vivo 10 medium (Invitrogen) to shortly stimulate cells prior to staining.

Cells used for immediate staining were incubated with antibodies against surface markers (CD45, CD4, CD8a, NKp46, and GARP) in the presence of a viability dye (eBioscience) and anti-CD16/32 to block FcγRs, fixed and permeabilized using FOXP3/ Transcription Factor Staining Buffer set (eBioscience) during 20 h at 4 °C, then stained with anti-FOXP3 in the presence of anti-CD16/32.

Cells used for ex vivo stimulations were incubated at 37 °C during 4 h in the presence or absence (no stimulation controls) of AH1 peptide (SPSYVYHQF, 10 μM), or PMA and Ionomycin (both from Sigma and at 500 ng/ml). Brefeldin A (Sigma, 5 μg/ml), anti-CD107a coupled to BV421 and the H2-L$^d$/AH1 tetramer coupled to PE (synthesized in-house) were added to the stimulation mix. After stimulation, cells were stained with antibodies against surface markers (CD16/CD32, CD45, CD4, and CD8a) in the presence of a viability dye (eBioscience) and anti-CD16/32 to block FcγRs, fixed and permeabilized with the Cytofix/Cytoperm kit (BD Biosciences), then stained with antibodies against intracellular cytokines (IFNγ and TNFα) in the presence of anti-CD16/32. Analyses were performed on a FACS LSR Fortessa flow cytometer (DIVA, BD Biosciences) and data were computed using the FlowJo software (Tree Star). Representative examples of gating strategies are illustrated in Supplementary Fig. 13.

**RT-qPCR and RNAseq of mouse tissue samples.** Mouse spleen and tumor fragments were collected and stored at −80 °C until processing. After tissue disruption with the Tissue Lyser (Quiagen), total RNA was extracted using Nucleospin Mini Columns (Macherey Nagel).

For RT-qPCR, RNA was reverse transcribed with Maxima First Strand cDNA Synthesis Kit (Thermofisher). qPCR was performed in a StepOnePlus device (Thermofisher) in reaction volumes of 20 μl containing 0.025 U/μl of Takyon Master Mix (Eurogentec), 300 nM of each primer, and 100 nM of Takyon probe under either standard conditions (95 °C for 3′; 45 cycles of 95 °C for 10″ and 60 °C for 30″) or fast conditions (95 °C for 3′; 95 °C for 3″ and 60 °C for 30″) depending on amplicon size. Sequences of primers and probes are listed in Supplementary Table 1.

For RNAseq, RNA quality was verified with an Agilent Bioanalyser. Sequencing was outsourced to Macrogen and performed on an Illumina Hiseq2000 with the following requirement: 100 pb of read length, paired-end reads, and 30 M reads/sample. Sequences were aligned to the GRCm38 mouse genome using the STAR software, and read counts were determined with the HTSeqCount software. Raw read counts were normalized using the DESeq2 package[42] and the R software.

**Human samples for RNAseq and multiplexed immunofluorescence.** Tumors and healthy tissues were obtained as surgical discard samples, or as research-aimed surgery or biopsy, after informed consent and under approval of the Commission d'Ethique Biomédicale Hospitalo-Facultaire, Brussels, Belgium (reference CEHF 2014/457).

**RNA extraction and RNAseq of human tissue samples.** Total RNA was extracted from sequential 10–30 μm cryosections cut from frozen tumor tissue, using the guanidinium isothiocyanate/cesium chloride procedure[43]. RNA quality was verified with an Agilent Bioanalyzer. A short-insert cDNA library was prepared with the TruSeq RNA-seq library kit (Illumina), and sequenced as 100 nucleotide-long paired-end reads using a HiSeq2000 sequencer (Illumina, ≥108 reads per sample). The read sequences were aligned to the GRCh38 human genome

sequence using Hisat2, and gene expression levels, normalized according to the FPKM method, were obtained with StringTie[44].

**Gene set enrichment analyses.** Hallmark gene signatures of response to IFNγ, TNFα, IL-2, or inflammation were downloaded from the MSigData base[30,32].

Experimental gene signatures were established in our laboratory using expression microarray data from various human primary or tumor cell lines exposed to cytokines in vitro. Melanoma cell lines LB2259-MEL.A and LB2667-MEL were established from the resected metastasis of two melanoma patients, and grown in IMDM medium (Invitrogen Life technologies) supplemented with 10% FCS and amino acid mix; 100 U/mL penicillin and 100 μg/mL streptomycin. Primary keratinocytes were derived from the detached epidermis layer of a healthy skin sample and were cultured in selective CnT-02 growth medium (Celltech). Dermal fibroblasts were obtained by mincing the dermal part of the skin sample and growing the cells in the same medium as the melanoma cell lines. Primary melanocytes and dermal endothelial cells (Promocell) were amplified in Medium 254 + HMGS2 (ThermoFischer Scientific) and Endothelial Cell Growth Medium (Promocell), respectively. The cultured cells were treated with either IFNγ (100 U/mL), IL-1β (10 ng/ml), TGF-β1 (5 ng/ml), TNFα (10 ng/ml), or no cytokine for 24 h. Total RNA was extracted and microarray assays were performed according to the Affymetrix Genechip Expression Analysis manual. Data acquisition and processing were conducted with Affymetrix Genechips Operating Software and Microsoft Excel. An additional normalization step was carried out by setting the mean expression level of each sample to 10 (arbitrary units). Expression data from a human CD4$^+$ T cell clone treated or not with TGF-β1 were obtained from GEO Series accession number GSE14330. A gene was considered induced by a given cytokine and included in the corresponding experimental gene signature if its level after cytokine treatment was >4 and was at least five times the level in the untreated condition in one of the five cell lines (experimental TGF-β1 signature in all cells), or in the CD4$^+$ T clone (experimental TGF-β1 signature in CD4$^+$ T cells). Genes of the experimental signatures are listed in Supplementary Data 2.

**Multiplexed immunofluorescence on mouse tumor sections.** Paraffin-embedded CT26 tumors were cut in 5-μm-thick sections then stained with anti-CD3 (CD3: Abcam #ab16669, clone SP7, diluted 1:500) and anti-CD8 (CD8: Cell signaling #98941, clone D4W2Z, diluted 1:400) primary antibodies. Ready-to-use Dako EnVision+ System-HRP Labelled Polymer Anti-Rabbit was used as detection reagent. Nuclei were stained with Hoescht reagent.

**Multiplexed immunofluorescence on human tumor sections.** Two adjacent 7-μm-thick cryosections freshly cut from frozen tonsil or tumor tissue were mounted on a microscope slide and immediately fixed in formaldehyde 4% during 5 min, washed with demineralized water (dH2O), and Tris-buffered saline (TBS), incubated with Peroxidase Blocking Reagent (Dako) for 10 min, washed with TBS supplemented with Tween20 (TBS-T), blocked with TBS + human immunoglobulins 1%, milk powder 2%, bovine serum albumin (BSA) 5% and Tween20 0.1% for 30 min, incubated with 5 μg/ml of mouse monoclonal anti-GARP antibody (clone MHG-6, in-house), in Dako Antibody Diluent or no primary antibody (negative control) for 90 min, incubated twice with TBS-T, incubated with secondary antibody (Dako EnVision + HRP Labelled Polymer Anti-mouse) for 60 min, washed twice with TBS-T, incubated with TSA Working Solution (PerkinElmer) + fluorescein dye 1:200 + H2O2 0.003% for 10 min, washed thrice with TBS-T, heated in citrate buffer pH6 in a microwave oven (3 min at 900 W and 15 min at 90 W) and washed twice with dH2O. The sections were kept immersed in dH2O until the next day. The slides were washed with TBS and underwent the same steps as above starting from the blocking step, with mouse monoclonal anti-CD34 antibody (clone Qbend 10, Abcam) diluted 1:200 as primary antibody, and BDP FL dye 1:200 instead of fluorescein. This second staining was omitted for the staining of the tumor samples. The slides were washed with TBS and underwent the same steps as above starting from the blocking step, with rabbit monoclonal anti-FOXP3 antibody (clone D608R, Cell Signaling Technology) diluted 1:100 as primary antibody, and Cy5 dye 1:200 instead of fluorescein. Finally, the slides were incubated with nuclear fluorescent dye Hoechst33258 1:1000 in TBS-T+10% BSA for 5 min, washed once in TBS-T and once in dH2O, mounted with Fluorescent Mounting Medium (Dako) and covered.

**Slide scanning and image analysis.** Digital 3 or 4-color images of the stained tissue sections were acquired with a Pannoramic P250 Flash III scanner (3DHistech) equipped with a Plan-Apochromat ×20/N.A. ×0.8 objective (Carl Zeiss) and a Point Grey Grasshopper 5MP camera, using DAPI1, FITC, SpRed, and Cy5 filter sets (Semrock). Images were analyzed with the Halo software and its Highplex FL, Area Quantification FL, and Cytonuclear FL modules (Indicalabs). The total number of cells was determined by dividing the total surface of nuclei by the mean nucleus size. The total number of FOXP3$^+$ and FOXP3$^+$GARP$^+$ cells was obtained by automated counting of FOXP3$^+$ nuclei, and FOXP3$^+$ nuclei surrounded by GARP stain, respectively. Adequate selection of FOXP3$^+$ cells was verified manually over a range (minimum 3) of fields of view. Because automated selection of FOXP3$^+$GARP$^+$ cells was found to comprise many false positives (mainly FOXP3$^+$ cells in contact with GARP$^-$stained blood vessels or fibrotic septa), we

validated all or a random fraction of at least 100 cells by visual inspection. For each sample, the number of validated FOXP3$^+$GARP$^+$ cells was calculated as the total number of FOXP3$^+$GARP$^+$ cells obtained by automated counting times the proportion of validated cells. The proportion of FOXP3$^+$ and FOXP3$^+$GARP$^+$ cells was obtained by dividing the respective validated counts by the total number of cells.

**Statistical analyses**. With the exception of RNAseq data, all statistical analyses were performed using the JMP®Pro 14 software. Comparisons of measurements taken at a single time point were performed using a two-tailed, non-paired, non-parametric Wilcoxon test. Comparisons of measurements repeatedly taken on each mouse over time (i.e. longitudinal data) were made using a mixed effects model, with a repeated covariance structure of the type "compound symmetry with unequal variances", applied on log-transformed tumor volumes. This approach is recommended for repeated measures over time on individual mice[45,46]. Post-hoc Tukey's test was performed to adjust for multiple comparisons. Statistical analyses of survival data presented in Kaplan Meier plots were performed with a Wilcoxon test. Numbers of mice ($n$) in the various experimental groups are indicated in the corresponding figure legend.

For Gene Set Enrichment analysis of RNAseq data, false discovery rate (FDR) and enrichment score (ES) were computed with the GSEA 3.0 Java software. Pearson correlations shown in Table S1 were calculated with Microsoft Excel V16.

**Reporting summary**. Further information on research design is available in the Nature Research Life Sciences Reporting Summary linked to this article.

## Data availability

The RNAseq data have been deposited in Gene Expression Omnibus repository with the primary accession code GSE153239 and GSE153388. Mouse (https://www.ensembl.org/Mus_musculus/Info/Index) and human (https://www.ensembl.org/Homo_sapiens/Info/Index) genome sequences for analyses of RNAseq data were retrieved from Ensembl. Hallmark genes signatures used in Figs. 5 and 8 were obtained from the Molecular Signatures Database and can be downloaded from: https://www.gsea-msigdb.org/gsea/msigdb/cards/HALLMARK_INTERFERON_GAMMA_RESPONSE.html; https://www.gsea-msigdb.org/gsea/msigdb/cards/HALLMARK_INFLAMMATORY_RESPONSE.html; https://www.gsea-msigdb.org/gsea/msigdb/cards/HALLMARK_IL2_STAT5_SIGNALING.html; https://www.gsea-msigdb.org/gsea/msigdb/cards/HALLMARK_TGF_BETA_SIGNALING.html; https://www.gsea-msigdb.org/gsea/msigdb/cards/HALLMARK_TNFA_SIGNALING_VIA_NFKB.html; https://www.gsea-msigdb.org/gsea/msigdb/cards/HALLMARK_SPERMATOGENESIS.html). Microarray data used to define experimental gene signatures are accessible with the accession codes GSE14330 and GSE154558. All the other data supporting the findings of this study are available within the article and its supplementary information files and from the corresponding author upon reasonable request.

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

## Acknowledgements

We thank S. Depelchin and A. Sibille for editorial assistance, A. Collignon and A. Daumerie for technical assistance, and J. Stockis and C. Barjon for helpful discussions and technical contributions. We also thank A.Y. Rudensky and the Memorial Sloan Kettering Cancer Center for providing the *B6.129(Cg)-Foxp3$^{tm4(YFP/icre)Ayr/J}$* mouse strain and Derya Unutmaz and the New York University for the *Lrrc32$^{tm1.1Hfuj}$* strain. This work was supported by grants from the Fondation contre le Cancer (grant F/2016/837), from the European Research Council (ERC) under the European Union's Horizon 2020 research and innovation programme (grant TARG-SUP 682818), from the Actions de Recherche Concertées (grant 14/19-056), from the Fonds National de la Recherche Scientifique (PDR number T.0089.16), and from Région Wallonne (program WALinnov, project IMMUCAN, convention number 1610119). G.d.S. was supported by a FRIA fellowship (F.R.S.-FNRS), and C.B. by a Télévie grant.

## Author contributions

S.Lu. conceived the study. G.d.S., N.C., S.Li., C.B., O.B., S.Le., J.D., M.G., B.v.d.W., M.S., H.d.H., H.D., E.V., W.M., P.G.C., N.v.B., and S.Lu. analyzed the data. S.Lu., G.d.S. and N.v.B. wrote the manuscript and performed statistical analyses. G.d.S., N.C., C.B., E.V., and W.M. performed the animals studies. N.v.B. performed mIF imaging and human RNAseq. N.v.B. and G.d.S. did the GSEA analyses. G.D.B., L.M., I.V.D.W., B.v.d.W., H.d.H., M.S., S.Li., and C.B derived, produced, and characterized the blocking anti-murine GARP:TGF-β1 antibody.

## Competing interests

G.D.B., L.M., I.V.D.W., B.v.d.W., H.d.H., and M.S. are full-time employees of argenx BVBA. Patents pertaining to the results presented in the paper have been filed under the Patent Cooperation Treaty (International application Number PCT/IB2019/053753), with S.Lu., G.d.S., S.Li., B.v.d.W., H.d.H., P.G.C., and M.S. as inventors and UCLouvain and argenx as applicants. S.Lu. received research support from argenx and owns stock options in argenx. The remaining authors declare no competing interests.
