## [Peer Review File · Nature Communications]

Reviewers' comments:

Reviewer #1 (Remarks to the Author):

This is a resubmission of an article investigating the potential of effector-negative anti-GARP-Tgfb1 IgG2a antibodies in murine preclinical immunotherapeutic oncology models. The manuscript shows that anti-GARP:TGFβ1 has a similar efficacy to that of anti-pan-TGFβ ligand antibodies in potentiating anti-PD-1 activity, albeit that the increase in efficacy is from only 2/10 complete responders with anti-PD-1 monotherapy versus 4/10 for anti-PD-1 combined with the anti-GARP:TGFβ1 drug. This is a small differential that is unlikely to be significantly different.

The authors have responded to both reviewers' requests on the statistical analysis of growth curves by stating that they used an analysis that takes into account longitudinal growth data. However, they still miss the point made by the reviewers, that there is missing data that artificially flattens the average growth curves, particularly of those lines wherein larger tumors are removed from study. For example, in Figure 2, the "average" growth curve data goes out to 45 days, yet the mice with the largest tumors are removed from the study at 15 days, thus the "average" tumor size after 15 days will be distorted downwards, leading to an artificial flattening of the growth rate in those arms from which tumors are removed. Therefore, average growth rates may only be compared up to the time when the first mouse reaches its endpoint (of excessive tumor size). A standard and alternative way of dealing with this type of data is to plot Kaplan Meyer survival graphs and undertake statistical analysis using a Mann–Whitney U test or Wilcoxon rank-sum test.

From data presented in Figure S2, it is impossible to tell how many tumors, from each arm of each of the five experiments displayed, shows complete tumor regression. The reader is therefore left with the impression that, whereas anti-PD-1 FcS monotherapy results in 2 to 3 complete tumor regressions per 10 mice, combination with anti-GARP:Tgfb1 FcS only raises this efficacy to 4 per ten mice (Main Fig 2). [Note that in main Fig. 2 there is a third tumor in the anti-PD-1 FcS monotherapy arm that is showing a downward trajectory in growth at termination, which on a Kaplan Meyer graph would be considered a complete response or censoring. I suggest that the KM plots and/or spider plots are shown in place of Fig S2. This may then also offer the opportunity for a meta-analysis of all five experiments, looking at the fraction of tumors that show complete responses. Indeed, the authors could also include enumeration of partial responders versus progressors. As most of the figures stand now, showing flawed "average growth curves", and no spider plots or KM survival plots, it is impossible to make this assessment. Figures S3 and S4 suffer from a similar problem, and KM graphs should replace average growth curves.

The authors make the claim that anti-GARP:TGF β 1 acts via effects in blunting the immunosuppressive activity of Tregs, however they do not provide any mechanistic data proving this to be the case. IN fact GARP is also clearly expressed at high levels in endothelial cells. The authors argument for not investigating drug responses in Treg-specific Garp-KO mice is not solid. They responded that, others have previously reported that Treg-specific Garp-KO mice do not show reduced growth of GL261 or MC38 tumors by comparison to WT. However, the correct experiment to undertake to address this point would be to treat tumor-bearing Foxp3-Garp-KO mice with anti-PD-1 in order to determine whether loss of Treg-associated GARP can synergize with anti-PD-1 in promoting tumor regression. I agree that it would take too long now to undertake the genetic KOs experiments for the purpose of this manuscript. An alternative strategy to address whether GARP expression on other cell types, such as endothelial cells, contribute to tumor regression, would be to immunologically deplete Tregs by, for example, using anti-CD25 depleting antibodies, and then determine whether anti-GARP-TGF β 1 has any additional effect on tumor growth (see).

Figure 2: what is the X axis of the middle column of graphs shown on the far left of the page? This is unnecessary, and makes it appear as though there is missing data.

Another unusual way that the authors have presented data is seen Figure 4. Here they present the number of cells per mm³, whereas it is more usual to present flow cytometry data as a percentage of live cells, or as a percentage of another cellular population. How do they calculate this parameter?

This issue also highlights the fact that the authors do not show immunohistochemical analysis/confirmation of their flow cytometry findings of no change in T cell numbers after therapy. Simple IHC for CD3+ or CD8+ T cells would support the flow cytometric findings. Moreover, it would also be informative to see the distribution of T cells across the tumor, especially in view of publications by Mariathasan et al., Nature 2018 and Dodagatta-Marri et al 2019 JITC, that show a redistribution of immune cells within the tumor after anti-TGF β therapy. This is pertinent to the current manuscript, since Dodagatta-Marri et al suggest that a considerable portion of the anti-tumor effect of anti-TGF β 1/2/3 therapy is due to drug-action on Treg cells.

On page 10, the authors highlight the difference between their Treg-mediated mechanism of blocking GARP:TGF β 1 with that of an anti-TGF β study in a mammary tumor model that exhibits spatial T cell exclusion from the tumor parenchyma, due to physical blockade by TGF β -responsive CAFs (Mariathasan et al. 2018). But their proposed mechanism of action of anti-GARP may be more similar to that described by Dodagatta-Marri et al 2019 for anti-TGF β 1-3 antibodies, i.e. blocking active TGF β on Tregs. Despite high GARP on endothelial cells, the author did not address the effect of anti-GARP:TGF β 1 on the vascular system. Are their morphological or functional changes in the vascular or lymphatic networks in vivo? This could be addressed for example by IHC. They mention unpublished negative data on in vitro analysis of GARP function on primary ECs, but this might not reflect the in vivo situation.

It is a shame that for immune profiling analysis of tumors, the authors only included 5 tumors per arm, and used the parameter of cells per mm² (Fig 4). It seems that there could a 5 or even 10 fold increase in CD4+ and CD8+ T cells in SOME tumors treated with anti-GARP:Tgfb1 (Figure 4a).

However, this effect is lost, statistically, by mixing data from responding and non-responding tumors in a population wherein only 40% of tumors respond.

How do the authors explain the increase in Tregs, particularly GARP+ Tregs, after treatment with anti-GARP:TGF β 1-FcD? Could this be due to increased proliferation of immature Tregs? Ki67 staining might address this. Can the authors isolate these Tregs and show that they are functionally less immunosuppressive, compared to those isolated from control-treated tumors? or at least stain use cellular markers indicative of Treg differentiation?

In two places, the authors refer to “Foxp3+ non-Treg T cells” but provide no citation to this cell type. Can the authors provide the reference or delete the statement.

Reviewer #2 (Remarks to the Author):

The aim of this study was to analyze the effect of a selective blockade of TGF- β 1 production by activated Tregs with antibodies against GARP:TGF- β 1 in tumor models. Overall the authors found that this complexes is sufficient to induce regressions of mouse tumors otherwise resistant to anti-PD1 immunotherapy. Mechanistically they linked this with increased effector functions of anti-tumor CD8+ T cells without increasing immune cell infiltration or depleting Tregs within tumors. Finally they found GARP-expressing Tregs in about one third of human cutaneous melanoma metastases. Thus, this study builds the basis for a novel therapy using anti-GARP:TGF- β 1 mAbs to overcome resistance to PD-1/PD-L1 blockade in patients with cancer.

Overall the authors have responded well to the reviewer comments. Specifically they performed additional experiments to show that their data are robust, generalizable, and statistically significant in multiple mouse tumor models. Thus, in my opinion this study does provide intriguing preclinical rationale for targeting this pathway for the immunotherapy of human cancers.

I agree with Reviewer 1 that this study lacks detailed mechanistic insights in vivo. But, I feel that the required experiments would go beyond the scope of this manuscript. However, these limitations should be discussed in this manuscript. This is especially the case for point 4 raised by reviewer 1.

Point by point reply to reviewer comments.

We would like to thank the reviewers for their positive reception of our work and their constructive input. In the following pages, each reviewer remark appears *verbatim* in black font, with our response immediately below in blue italic font.

Reviewer #1 (Remarks to the Author):

This is a resubmission of an article investigating the potential of effector-negative anti-GARP-Tgfb1 IgG2a antibodies in murine preclinical immunotherapeutic oncology models. The manuscript shows that anti-GARP: TGFB1 has a similar efficacy to that of anti-pan-TGF β ligand antibodies in potentiating anti-PD-1 activity, albeit that the increase in efficacy is from only 2/10 complete responders with anti-PD-1 monotherapy versus 4/10 for anti-PD-1 combined with the anti-GARP: TGFB1 drug. This is a small differential that is unlikely to be significantly different.

The reviewer cites partial results from Fig. 2, comparing 2 groups from an experiment which included a total of 10 groups, ignoring the 10 other independent tumor experiments shown in the manuscript, and concluding that this small difference is unlikely to be significant. Our reply:

- *Notwithstanding the other groups and experiments, this difference alone corresponds to a doubling in the proportion of complete responders (CR) to combination therapy by comparison to anti-PD-1 monotherapy. This is not a small difference.*
- *Similar 2.8 to 5-fold increases in proportions of CR were observed in the 10 other experiments, and were explicitly illustrated on graphs of individual tumor growth curves in submitted Fig. 2, 5a-b, S3c, and S4b-c. We agree that this was probably less clearly illustrated in the 5 panels of submitted Fig. S2, which only showed average tumor growth curves, in an attempt to summarize the data. We have revised all figures in order to: i) include graphs showing individual tumor growth curves for all experiments (including experiments shown in former Fig. S2, now shown in revised Fig. 3); ii) use a log2 scale to report tumor size, as this better illustrates not only CR but also partial responders (PR), which are now clearly enumerated in response to another comment of the reviewer below.*
- *Importantly, we had reported exact P values for the statistical analyses of all 11 independent experiments. P values for differences between anti-PD-1 monotherapy versus anti-PD-1 combined with anti-GARP: TGF- β 1 are <0.05 in experiment shown in Fig.2, are <0.05 in 8 other independent experiments, and = 0.0685 and 0.0675 in the last 2 of 11 experiments. Finally, to address another comment of the reviewer below, we have used an additional statistical approach for the meta-analysis of data shown in revised Fig. 3. P values for differences between anti-PD-1 monotherapy versus anti-PD-1 combined with anti-GARP: TGF- β 1 in these meta-analyses are = 0.005. Our conclusion, unlike that of the reviewer, is that **it is highly unlikely (<5% probability) that there is no difference between the treatments.***

The authors have responded to both reviewers' requests on the statistical analysis of growth curves by stating that they used an analysis that takes into account longitudinal growth data. However, they still miss the point made by the reviewers, that there is missing data that artificially flattens the average growth curves, particularly of those lines wherein larger tumors are removed from study.

There is no missing data in the figures, as larger tumors are not removed from the studies to calculate average tumor sizes. The average growth curves are therefore not artificially flattened by missing data, in Fig. 2 or anywhere. Although this can be deduced from the graphs showing tumor growth curves in individual mice, we agree with the reviewer that it could be more clearly explained yet. We have thus added the following paragraph in "Material and Methods":

"Once a tumor reaches the maximum tolerated size (pre-defined as an ethical humane endpoint in our protocol), the corresponding mouse was euthanized. The last tumor size measured prior to euthanasia (i.e. maximum size) is carried forward in the data series for later time points. Average

tumor size per group is calculated for all tumors, including the largest tumors, at all time points. As soon as all mice in a given group have been euthanized, the average size is not computed anymore for that group, and the average growth curve is interrupted on the graphs.”

This clarifies better that curve flattening is not artificially induced by tumors being removed from study, but instead is exclusively due to CR and PR surviving long term in groups treated with combinations of antibodies. It also clarifies our reply to a point made by one reviewer of our original manuscript submitted to Nature Medicine. The point was: “ ... it is unclear what was done with mice that reached criteria...”. These mice are euthanized, as explained in M&M, but we had failed to understand this point as a suggestion that there was missing data in our figures.

For example, in Figure 2, the “average” growth curve data goes out to 45 days, yet the mice with the largest tumors are removed from the study at 15 days, thus the “average” tumor size after 15 days will be distorted downwards, leading to an artificial flattening of the growth rate in those arms from which tumors are removed.

Again, no mice are removed from the study to calculate average tumor size at any point in Fig. 2 or anywhere. When average tumor growth curves are “distorted downwards”, it is because tumors start to regress and will be completely rejected in some mice, as can be seen on graphs of individual tumor growth shown on all revised figures.

Therefore, average growth rates may only be compared up to the time when the first mouse reaches its endpoint (of excessive tumor size).

In Fig. 2, a first mouse reaches the endpoint on day 14. It belongs to the group receiving the isotype control antibody. Comparisons need to be continued after day 14 because i) treatments are still ongoing, and ii) complete responses (tumor rejections) in mice treated with antibody combinations occur in a vast majority of the cases after day 20. Complete responses, even if delayed by comparison to first death, are the most interesting and significant events observed in response to immunotherapies.

A standard and alternative way of dealing with this type of data is to plot Kaplan Meyer survival graphs and undertake statistical analysis using a Mann–Whitney U test or Wilcoxon rank-sum test.

We agree that Kaplan Meyer (KM) survival graphs and log rank tests are an interesting alternative, particularly well-suited for analyses of survival data derived from large numbers of patients or mice. In Fig. 2, the relatively low number of mice per group (n=9-10) precludes use of this approach for this given single experiment. This experiment is nevertheless crucial to our manuscript because it contains several important control groups that are not present in the other confirmatory experiments shown later in the manuscript.

To address this comment and another one below, we propose to use KM representation for meta-analyses of data pooled from 7 experiments comparing anti-PD-1 alone to anti-PD-1 combined with anti-GARP:TGF- β 1, and to include these graphs in a new main figure (revised Fig. 3) which would replace former Supplementary Fig. S2. Groups in the meta-analyses comprise 29-39 mice, which allows for KM plots and statistical analysis using a Wilcoxon test. P values for differences in survival between anti-PD-1 combined with anti-GARP:TGF- β 1 and anti-PD-1 alone are 0.005, whether the antibodies are used as WT antibodies or as Fc-dead variants. We thank reviewer #1 for this excellent suggestion, as this allows us to demonstrate our point more clearly on lines 130-131 of the revised manuscript.

From data presented in Figure S2, it is impossible to tell how many tumors, from each arm of each of the five experiments displayed, shows complete tumor regression.

We agree that this was impossible to tell in the original Supplementary Fig. S2, because in contrast to all other figures, it did not show individual tumor growth curves. These curves are now shown in revised Fig. 3, which replaces former Fig. S2 and indicates proportions of not only

complete tumor regressions or responses (CR), but also partial responses (PR, see other comment below). Due to the large amount of data and to follow another suggestion of the reviewer, revised Fig. 3 is presented as 2 meta-analyses of a total of 7 experiments (4 pooled experiments with WT antibodies, 3 pooled experiments with Fc dead antibodies).

The reader is therefore left with the impression that, whereas anti-PD-1 FcS monotherapy results in 2 to 3 complete tumor regressions per 10 mice, combination with anti-GARP:Tgfb1 FcS only raises this efficacy to 4 per ten mice (Main Fig 2).

This comment relates to partial results from one experiment shown in Fig. 2, but fails to acknowledge results from 10 other independent experiments. It has been mostly addressed above, except for the reference to the “third” tumor regression, which we address below.

[Note that in main Fig. 2 there is a third tumor in the anti-PD-1 FcS monotherapy arm that is showing a downward trajectory in growth at termination, which on a Kaplan Meyer graph would be considered a complete response or censoring.

We define complete responders (CR) as mice alive with no detectable tumor at the end of the experiment. Thus, this third tumor can be considered, at best, as a partial response: it shows a “downward trajectory”, but persists until the end of the experiment with a volume $> 256 \text{ mm}^3$, i.e. largely above the detection limit ($\pm 32 \text{ mm}^3$). The mouse never reaches the ethical humane endpoint and survives until the experiment is terminated on day 45.

We had not highlighted nor discussed partial responses in our submitted manuscript. We agree with the reviewer that these responses can also be interesting and informative... as long as they are reported for all groups! Thus, revised figures now all indicate proportions of partial responders (PR), defined as mice that are alive on day 35 or 40 (depending on the tumor model) and bear a tumor $>32 \text{ mm}^3$ at the time of euthanasia. Day 35 or 40 are arbitrary time points at which all mice receiving the isotype control antibody have reached the ethical humane endpoint in a given experiment. Definitions of CR and PR are indicated in the revised figure legends.

Revised main Fig. 2, for example, shows 2 CR + 1 PR in mice treated with anti-PD-1 FcS monotherapy (green lines in Fig 2; $n=10$), and 4 CR + 2 PR in mice treated with anti-PD-1 FcS combined with anti-GARP:TGF- β 1 FcD (red dotted lines; $n=10$). Proportions of PR, like that of CR, are approximately doubled in the combination arm by comparison to monotherapy. Similar observations are made in all experiments.

I suggest that the KM plots and/or spider plots are shown in place of Fig S2. This may then also offer the opportunity for a meta-analysis of all five experiments, looking at the fraction of tumors that show complete responses. Indeed, the authors could also include enumeration of partial responders versus progressors. As most of the figures stand now, showing flawed “average growth curves”, and no spider plots or KM survival plots, it is impossible to make this assessment. Figures S3 and S4 suffer from a similar problem, and KM graphs should replace average growth curves.

We believe this point has been fully addressed in reply to other the points above: i) we show KM plots for the meta-analyses of 7 experiments in revised Fig. 3; and ii) we show individual tumor growth curves including enumeration of proportions of CR and PR in all figures, including former Fig. S3 and S4 (i.e. revised Fig. S2 and S3). Please note that former Fig. S3 and S4 already showed individual growth curves.

To represent individual tumor growth curves, we have chosen a Log2 scale for tumor volumes on the Y axis, instead of percent change relative to baseline as used in “spider plots”. Although we agree with the reviewer that spider plots are very useful to represent evolution of tumor burden in patients with cancer, they are not well suited for the analyses of mouse tumors. In contrast, a Log2 scale of tumor volume illustrates very clearly the difference between progressors (or non-responders, NR), PR and CR.

Altogether, these changes in data representation greatly improve our manuscript, and we thank reviewer #1 for these excellent suggestions.

Additional note to justify not using spider plots: in standard transplantable mouse tumor experiments, treatment is started on day 6 after tumor cell inoculation, when tumors are detectable and measurable in all mice but are still small by comparison to their size at the ethical humane endpoint. Consequently, whereas CR undergo 100% reduction in tumor volume relative to baseline, progressors experience up to 4000% increase (e.g. progression from a small detectable tumor of $\pm 32 \text{ mm}^3$ to a maximum of $\pm 1300 \text{ mm}^3$). Spider plots on mouse data yield graphs that are very difficult to read and interpret. We provide an example below, in which a spider plot is used to illustrate partial data shown in revised main Fig. 2.

To the best of our knowledge, spider plots are very well suited to plot evolution of tumor volumes in patients, but unfortunately not in mouse tumor models.

The authors make the claim that anti-GARP:TGFβ1 acts via effects in blunting the immunosuppressive activity of Tregs, however they do not provide any mechanistic data proving this to be the case. IN fact GARP is also clearly expressed at high levels in endothelial cells. The authors argument for not investigating drug responses in Treg-specific Garp-KO mice is not solid. They responded that, others have previously reported that Treg-specific Garp-KO mice do not show reduced growth of GL261 or MC38 tumors by comparison to WT.

This was only part of our response, meant mainly to cite the only available published data relevant to this question. But this argument was not sufficient, we agree, and we had thus suggested other experiments (see below).

However, the correct experiment to undertake to address this point would be to treat tumor-bearing Foxp3-Garp-KO mice with anti-PD-1 in order to determine whether loss of Treg-associated GARP can synergize with anti-PD-1 in promoting tumor regression. I agree that it would take too long now to undertake the genetic KOs experiments for the purpose of this manuscript.

We had answered previously that we were considering performing experiments in Foxp3-Garp-KO mice (Foxp3^{Cre} x GARP^{fl/fl} mice), but to us, the “correct” experiments should:

- *Include groups of GARP^{fl/fl} mice with a GARP deletion not only in Tregs (Foxp3^{Cre}), but also in platelets (Pf4^{Cre}), B cells (Mb1^{Cre}) or endothelial cells (Tie2^{Cre}), i.e. in the main cell types which have been shown to express GARP;*
- *Treat tumor-bearing mice not only with anti-PD-1 alone, but also with anti-PD-1 combined with anti-GARP:TGF-β1.*

Altogether, these experiments should allow to determine in which cell-specific KO strain i) anti-PD-1 alone exerts superior anti-tumor efficacy than in WT littermates; and ii) combination of anti-PD-1 with anti-GARP:TGF-β1 is not superior to anti-PD-1 alone, in contrast to WT littermates. Superior efficacy of anti-PD-1 alone in KO vs WT will not necessarily be observed in any of the strains because cells with a genetic deletion of GARP may acquire or enhance compensatory immunosuppressive mechanisms during their development or differentiation in the absence of GARP. In contrast, absence of superior efficacy of anti-GARP combined with anti-

PD-1 by comparison to anti-PD-1 alone will identify which cells need to be targeted by anti-GARP:TGF- β 1 antibodies to observe anti-tumor efficacy.

*When responding to the first round of reviews (received in July 2019), we had estimated that we would need 18-24 months to perform the experiments in 8 different genetically modified mouse strains (the original reviewer was asking for cell-specific conditional KO of *Garp* and *Tgfb1*). We had already undertaken the establishment of 4 mouse strains (*Garp* KOs).*

*To date, we have been able to perform the tumor experiments in 2 strains, namely in *Foxp3^{Cre} x Garp^{fl/fl}* and *Pf4^{Cre} x Garp^{fl/fl}* mice, which is more than what the current reviewer is suggesting.*

We have included these important results as a new main figure (revised Fig. 7) described on lines 273-294 in the revised manuscript. We can now firmly conclude that targeting GARP:TGF- β 1 on Tregs, but not on platelets, is necessary to increase the anti-tumor activity of anti-PD-1. Our results also imply that targeting GARP:TGF- β 1 on endothelial cells is not sufficient to observe anti-tumor activity.

We believe that these new data are the most important part of our reply to reviewers. They firmly establish the mechanism and specificity of anti-tumor activity of anti-GARP:TGF- β 1 mAbs, which are now clearly demonstrated to act by targeting Tregs.

An alternative strategy to address whether GARP expression on other cell types, such as endothelial cells, contribute to tumor regression, would be to immunologically deplete Tregs by, for example, using anti-CD25 depleting antibodies, and then determine whether anti-GARP-TGFB1 has any additional effect on tumor growth (see).

*Our new data in Treg-specific *Garp* KO mice clearly establish that it is necessary to target GARP on Tregs but not sufficient to target GARP on endothelial cells to observe anti-tumor activity with anti-GARP:TGF- β 1 mAbs.*

Notwithstanding this, we considered also the alternative strategy proposed by reviewer #1, but unfortunately had to conclude that it is not technically feasible with the currently available anti-CD25 antibodies, including with variants of the most widely used anti-CD25 clone PC-61. This clone has been used under its original format (rat IgG1, commercially available), or under a format with reduced binding to inhibitory Fc γ R2b (mouse IgG2a variant constructed and described by Arce-Vargas et al, Immunity 2017; variant not commercially available).

Technical limitations are the following:

- *When used as a rat IgG1, anti-CD25 PC-61 partially depletes Tregs in the blood and peripheral lymphoid organs, but not within tumors because of high intra-tumoral expression of inhibitory Fc γ R2b (Arce-Vargas et al, 2017, Immunity). This explains why anti-CD25 PC-61 rIgG1 exerts anti-tumor activity when administered in a “prophylactic” setting, i.e. before or soon after tumor challenge (Golgher et al., 2002; Jones et al., 2002; Onizuka et al., 1999; Quezada et al., 2008; Shimizu et al., 1999), but not when administered in a “therapeutic” setting, i.e. when mice carry established tumors (Golgher et al., 2002; Jones et al., 2002; Onizuka et al., 1999; Shimizu et al., 1999). When we administered anti-CD25 PC-61 rIgG1 to mice before transplantation of CT26 tumor cells, we observed tumor rejections and survival of 100% of the mice (data not shown). This high anti-tumor activity of anti-CD25 PC-61 rIgG1 in a prophylactic setting leaves no room to examine additional effects of anti-GARP:TGF- β 1 on CT26 tumor growth.*
- *When used as a mouse IgG2a in a therapeutic setting, anti-CD25 PC-61 i) is able to deplete Tregs within tumors; ii) exerts anti-tumor activity; and iii) induces survival of 100% of CT26 tumor-bearing mice when combined with anti-PD-1 (Arce-Vargas et al, 2017, Immunity). Again, notwithstanding the fact that we do not have access to this antibody variant, the high anti-tumor activity of the anti-CD25 PC-61 mIgG2a variant in a therapeutic setting leaves no room to examine additional effects of anti-GARP:TGF- β 1 on CT26 tumor growth.*

Yet another strategy to deplete Tregs consists in administering diphtheria toxin (DT) to C57BL/6 mice carrying a human Diphtheria Toxin Receptor (DTR) transgene in the *Foxp3* locus (*Foxp3^{DTR}* mice). Administration of DT to MC38 tumor-bearing mice as late as 7 days after tumor cell transplantation depletes >95% of the Tregs. But again, Treg depletion in this therapeutic setting exerts very potent anti-tumor activity (100% survival), leaving no room to examine additional effects of anti-GARP:TGF- β 1 on MC38 tumor growth (see our data in the figure for reviewers below).

Figure 2: what is the X axis of the middle column of graphs shown on the far left of the page? This is unnecessary, and makes it appear as though there is missing data.

As shown at the bottom left of panel b, all X axes indicate “Days” after tumor cell transplantation and all Y axes indicate “Tumor volume”. X and Y axes have identical scales and tick marks in all graphs, with numbers indicated for X axes below the most bottom graph in each column, and for Y axes next to the far left graphs in each row.

We do not understand what the reviewer means by “middle column of graphs on the far left of the page” ...

We do not understand either what exactly is “unnecessary”.

Finally, there is of course no missing data, as already explained above.

Could we call upon the editor for help in interpreting the reviewer comment in this particular case? We are more than willing to make our figure as clear as possible.

Another unusual way that the authors have presented data is seen Figure 4. Here they present the number of cells per mm³, whereas it is more usual to present flow cytometry data as a percentage of live cells, or as a percentage of another cellular population. How do they calculate this parameter?

It is not unusual to represent numbers of tumor infiltrating leukocytes (TILs) per mm³ or per g of tumor. This parameter corresponds to the density of TILs, i.e. to numbers of TILs normalized by the size of the tumor. Here are a few selected references of publications in high ranking journals using this parameter (number/g): Martin et al., Science Translational Medicine, 2020, Lan et al. Science Translational Medicine, 2018 ; Wang et al., Nature Communication, 2018 ; Williford, et al. Science Advances, 2019; ...

We calculate densities of TILs (and TIL subsets) as follows:

- *total number of TILs = percentage of TILs within live cells x the total number of cells isolated from the tumor. The percentage of TILs (and TIL subsets) within live cells is determined by flow cytometry, and the number of live cells isolated from the tumor is*

counted after tumor collection and mechanical dissociation using an automated cell counter that discriminates live and dead cells with a dead cell marker. A sentence has been added to Material and Methods to clarify this point (lines 481-482).

- *total number of TILs / mm³ of tumor = total number of TILs / tumor volume. The latter is calculated from large and small tumor diameters measured with a caliper, as indicated in the Material and Methods. We could of course also represent numbers of TILs/g. We have also calculated this parameter, and the conclusions are the same, as expected.*

As TIL numbers / mm³ of tumor are calculated from flow cytometry data, it is of course also possible to represent percentages of TILs within live cells or another cell population (such as CD45⁺ cells). We were already citing some of these percentages in the text of the submitted manuscript.

To address the referee comment, we have revised our manuscript as follows:

- *We have added a new panel to revised Fig. 5 (i.e. original Fig. 4) to indicate the volumes (mm³) and weights (g) of the tumors at the time of euthanasia on day 13.*
- *We have added a new Supplementary Figure S5 which graphically illustrates percentages of TILs and TIL subsets, in addition to TIL densities shown in revised Fig. 5. This alternative representation of our data, requested by the reviewer, does not change our conclusions, as expected.*

This issue also highlights the fact that the authors do not show immunohistochemical analysis/confirmation of their flow cytometry findings of no change in T cell numbers after therapy. Simple IHC for CD3+ or CD8+ T cells would support the flow cytometric findings. Moreover, it would also be informative to see the distribution of T cells across the tumor, especially in view of publications by Mariathasan et al., Nature 2018 and Dodagatta-Marri et al 2019 JITC, that show a redistribution of immune cells within the tumor after anti-TGF β therapy. This is pertinent to the current manuscript, since Dodagatta-Marri et al suggest that a considerable portion of the anti-tumor effect of anti-TGF β 1/2/3 therapy is due to drug-action on Treg cells.

We have performed a new CT26 tumor experiment to analyze T cell numbers and distribution within tumor sections by immunofluorescence microscopy using anti-CD3 and anti-CD8 antibodies as suggested by the reviewer. These data are shown in a new Supplementary Fig. S7. They confirm flow cytometry findings shown in revised Fig. 5: there is no increase in total T, CD8 T and CD4 T cells numbers in tumors from mice treated with anti-GARP:TGF- β 1 combined with anti-PD-1 by comparison to anti-PD-1 alone. They also show that there is no difference in the distribution of T cells between the periphery and center of the tumor in mice treated with anti-GARP:TGF- β 1 combined with anti-PD-1 by comparison to anti-PD-1 alone. These results are described on lines 187-193 of the revised manuscript as:

“We also estimated densities and distribution of CD4 and CD8 T cells in formalin-fixed paraffin-embedded (FFPE) tumor sections by immunofluorescence microscopy and quantitative digital imaging. Here again, we observed no significant difference between the various treatment groups (Supplementary Fig. 7a-c). A statistically non-significant trend towards increased densities of T cells, apparent in both the periphery and center of tumors, was observed in mice treated with anti-PD-1, whether it was combined or not with anti-GARP:TGF- β 1 (Supplementary Fig. 7c).”

On page 10, the authors highlight the difference between their Treg-mediated mechanism of blocking GARP:TGF β 1 with that of an anti-TGF β study in a mammary tumor model that exhibits spatial T cell exclusion from the tumor parenchyma, due to physical blockade by TGF β -responsive CAFs (Mariathasan et al. 2018). But their proposed mechanism of action of anti-GARP may be more similar to that described by Dodagatta-Marri et al 2019 for anti-TGF β 1-3 antibodies, i.e. blocking active TGF β on Tregs.

Our hypothesis is that anti-GARP:TGF- β 1 mAbs exert anti-tumor effects by blocking active TGF- β 1 production by Tregs. It is thus different from the two mechanisms of action for anti-TGF- β mAbs proposed by Mariathasan et al and Dodagatta-Marri et al.

Our hypothesis for anti-GARP:TGF- β 1 mAbs is supported by the following data: i) anti-GARP:TGF- β 1 blocks active TGF- β 1 production by Tregs in vitro (Fig. 1); ii) it exerts anti-tumor activity when used as an Fc-dead variant, i.e. a variant able to block active TGF- β 1 production but not to kill GARP-expressing cells (Fig. 2, 3, 6, 7, S2, S3); iii) it exerts anti-tumor activity without depleting Tregs or reducing Treg densities within tumors (Fig. 5); and finally iv) it does not exert anti-tumor activity in mice carrying a Treg-specific deletion of Garp (Fig. 7).

Altogether, this shows that anti-GARP:TGF- β 1 works by blocking active TGF- β 1 production by Tregs.

This mechanism is different from the mechanism of action proposed by Mariathasan et al, who suggest that anti-TGF- β 1,2,3 blocks TGF β signaling in stromal cells and facilitates T-cell penetration into tumors. We do not observe increased penetration of T cells within tumors in mice treated with anti-GARP:TGF- β 1 (Fig. 5, S5-7).

It is also different from the mechanism of action of anti-TGF- β 1,2,3 proposed by Dodagatta-Marri et al, who suggest that anti-TGF- β combined with anti-PD-1 acts by blocking TGF- β activity in Tregs and tumor cells. This suggestion is based on observations that anti-TGF- β 1,2,3 reverts the increases in Treg/ Th ratios and phosphoSMAD3 in tumor cells that are induced by anti-PD-1 alone. We do not observe a reduction in Treg / Th ratio in mice treated with anti-GARP:TGF- β 1 (Fig. 5, S5-6).

We have modified the discussion to cite the Dogatta-Marri report, in addition to the Mariathasan et al report (lines 358-366 of the revised manuscript).

Despite high GARP on endothelial cells, the author did not address the effect of anti-GARP:TGF β 1 on the vascular system. Are their morphological or functional changes in the vascular or lymphatic networks in vivo? This could be addressed for example by IHC. They mention unpublished negative data on in vitro analysis of GARP function on primary ECs, but this might not reflect the in vivo situation.

As detailed above, our new data in Treg- and platelet-specific Garp KO mice clearly establish that whereas it is necessary to target GARP on Tregs, it is not necessary to target it on platelets and not sufficient to target it on endothelial cells to observe anti-tumor activity with anti-GARP:TGF- β 1 mAbs.

We did not observe gross morphological changes in CT26 and MC38 tumors from mice treated with any mAb or mAb combination used in this study. Examining morphological and functional changes in the vascular and lymphatic networks in tumor-bearing mice treated with anti-GARP:TGF- β 1 mAbs falls far beyond the scope of our manuscript. But we thank the reviewer for this interesting suggestion.

It is a shame that for immune profiling analysis of tumors, the authors only included 5 tumors per arm, and used the parameter of cells per mm² (Fig 4). It seems that there could a 5 or even 10 fold increase in CD4+ and CD8+ T cells in SOME tumors treated with anti-GARP:Tgfb1 (Figure 4a). However, this effect is lost, statistically, by mixing data from responding and non-responding tumors in a population wherein only 40% of tumors respond.

We discussed above our choice to represent densities of TILs (cell numbers per mm³) in revised Fig. 5. As requested by the reviewer, we now also illustrate proportions (%) of these cells in Supplementary Fig. S5, but this does not change our interpretation and conclusions.

Immune profiling in Fig. 5 was performed on tumors collected from 5 mice per group and a total of 8 treatment groups. This represents 40 individual tumors. The experiment was repeated twice with the most important groups. We now added results pooled from these repeat

experiments in a new Supplementary Fig. S6, which includes 10 additional mice per group. We still observe no statistically significant increase in CD4+ or CD8+ T cells.

Immune profiling can only be performed before tumors are completely rejected in some mice, to obtain sufficient tumor material for analyses of all mice (including future CR). This explains our choice of performing immune profiling on day 13 after tumor cell transplantation. At that time point however, it is impossible to predict with certainty which tumors would ultimately be completely rejected at a later time point. This is illustrated in the figure for the reviewer below, showing the evolution of individual tumor volumes until day 13 in experiment from Fig. 5.

We have added a panel to Fig. 5 and S7 to illustrate tumor weights and volumes on day 13. These panels show that there is already a statistically significant trend towards reduced tumor weights in groups treated with anti-GARP:TGF- β 1 combined with anti-PD-1 by comparison to controls (revised Fig. 5a and S7). However, no rigorous prediction can be made on the future CR, PR or NR status of the corresponding mice.

This is why we had rather chosen to show correlations between immune profiling parameters and tumor weights in our submitted manuscript. We had concluded that tumor weights inversely correlated with proportions of anti-AHI CD8⁺ T cells with multiple effector functions (revised Fig. 5g), but not with other parameters. We now added more precisely that tumor weights did not inversely correlate with densities of total leukocytes or any leukocyte subset, and show selected examples of correlation analyses in Supplementary Fig. S5b. This is described on lines 247-252 of the revised manuscript.

How do the authors explain the increase in Tregs, particularly GARP⁺ Tregs, after treatment with anti-GARP:TGFβ1-FcD? Could this be due to increased proliferation of immature Tregs? Ki67 staining might address this. Can the authors isolate these Tregs and show that they are functionally less immunosuppressive, compared to those isolated from control-treated tumors? or at least stain use cellular markers indicative of Treg differentiation?

As described on lines 206-208 of the revised manuscript, “numbers of total Tregs and GARP⁺ Tregs per mm³ of tumor were not decreased in mice that had received an anti-GARP:TGF-β1 mAb, alone or in combination with anti-PD-1 (Fig. 5c and Supplementary Fig. S6a). If anything, an increase in Treg and GARP⁺ Treg numbers was observed in mice treated with the anti-GARP:TGF-β1 FcD + anti-PD-1 combination in one experiment (Fig. 5c), but this was not confirmed in two others (Supplementary Fig. S6a).”

The important point is that Tregs are not depleted.

We have not isolated Tregs to test if they were less suppressive after treatment with anti-GARP:TGF-β1 FcD in vivo because this is technically not feasible (due to low numbers of cells that can be isolated from mouse tumors).

In two places, the authors refer to “Foxp3⁺ non-Treg T cells” but provide no citation to this cell type. Can the authors provide the reference or delete the statement.

In contrast to non-Treg mouse T cells, non-Treg human T cells frequently express FOXP3 in response to TCR stimulation. This is a notorious difference between human and mouse T cell biology, repeatedly reported by many groups since the discovery of FOXP3 and its role in Treg biology. We had provided one reference for this observation (reference 18), and we will not delete the statement.

Reviewer #2 (Remarks to the Author):

The aim of this study was to analyze the effect of a selective blockade of TGF-β1 production by activated Tregs with antibodies against GARP:TGF-β1 in tumor models. Overall the authors found that this complex is sufficient to induce regressions of mouse tumors otherwise resistant to anti-PD-1 immunotherapy. Mechanistically they linked this with increased effector functions of anti-tumor CD8⁺ T cells without increasing immune cell infiltration or depleting Tregs within tumors. Finally they found GARP-expressing Tregs in about one third of human cutaneous melanoma metastases. Thus, this study builds the basis for a novel therapy using anti-GARP:TGF-β1 mAbs to overcome resistance to PD-1/PD-L1 blockade in patients with cancer.

Overall the authors have responded well to the reviewer comments. Specifically they performed additional experiments to show that their data are robust, generalizable, and statistically significant in multiple mouse tumor models. Thus, in my opinion this study does provide intriguing preclinical rationale for targeting this pathway for the immunotherapy of human cancers.

I agree with Reviewer 1 that this study lacks detailed mechanistic insights in vivo. But, I feel that the required experiments would go beyond the scope of this manuscript. However, these limitations should

be discussed in this manuscript. This is especially the case for point 4 raised by reviewer 1.

We thank reviewer #2 for his support and comments on the robustness, generalizability and statistical significance of our data. We are also very pleased that we can now provide data in Treg-specific Garp KO mice, as requested in point 4 by reviewer #1 in Nature Medicine. It took us almost a year to obtain robust and reproducible results in tumor experiments with these mice (revised Fig. 7). These data unequivocally support our hypothesis regarding the mechanism of action of blocking anti:GARP:TGF- β 1 antibodies.

REVIEWERS' COMMENTS:

Reviewer #1 (Remarks to the Author):

The authors have more than adequately addressed the reviewers' comments, and should be congratulated on an excellent study. Their inclusion of tumor studies in cell specific KO of Garp in platelets or Tregs is commendable, but unfortunately no statistics were provided, and it is unlikely that the data in Figure 7b show any statistical significance between groups. This reviewer understands the problems of reaching statistical significance in studies of this type, but to accommodate this issue, the two paragraphs in Results and Discussion relating to Figure 7b should be modified as suggested below, where underlined text are changes to the original:

Line 286 to 290

As shown in Fig. 7b, the anti-tumor efficacy of anti-GARP:TGF- β 1 combined with anti-PD-1 trended towards better responses than to anti-PD-1 alone in both platelet-specific Garp KO mice (CR: 42% vs 23%) and their WT littermates (CR: 46% vs 13%), similar to observations in our previous experiments, although statistical significance was not reached. In contrast, Fig. 7c shows that anti-GARP:TGF- β 1 combined with anti-PD-1 showed the opposite trajectory to that seen in platelet-specific Garp KO mice or wild type mice. Anti-GARP:TGF- β 1 combined with anti-PD-1 in Treg-specific Garp KO mice trended towards inferior responses compared to anti-PD-1 alone (CR: 19% vs 28%), with the opposite trend observed in wild type litter mice (CR: 34% vs 20%).

Lines 352-353

Our experiments in cell-specific Garp KO mice (<establish>) suggest that blocking the activity of TGF- β 1 emanating from GARP-expressing Tregs is required for anti-GARP:TGF- β 1 to exert antitumor activity.

Line 353-354

Either delete the sentence: "Blocking the activity of TGF- β 1 emanating from GARP-expressing platelets is not necessary, and blocking that from other GARP-expressing cells, such as endothelial cells, is not sufficient"; or replace it with "Whether GARP-expressing platelets or GARP-expression from other cell types contributes to tumor rejection requires more extensive studies."

Reviewer #2 (Remarks to the Author):

The points raised in the previous round of review have been satisfactorily addressed.

Point by point reply to reviewers' comments.

We thank the referees for their very constructive input. Their comments appear below in black font, with our responses in blue font. We used tracked changes to highlight the corresponding changes in the manuscript Word file uploaded on the MTS.

REVIEWERS' COMMENTS:

Reviewer #1 (Remarks to the Author):

The authors have more than adequately addressed the reviewers' comments, and should be congratulated on an excellent study. Their inclusion of tumor studies in cell specific KO of Garp in platelets or Tregs is commendable, but unfortunately no statistics were provided, and it is unlikely that the data in Figure 7b show any statistical significance between groups. This reviewer understands the problems of reaching statistical significance in studies of this type, but to accommodate this issue, the two paragraphs in Results and Discussion relating to Figure 7b should be modified as suggested below, where underlined text are changes to the original:

Line 286 to 290

As shown in Fig. 7b, the anti-tumor efficacy of anti-GARP:TGF- β 1 combined with anti-PD-1 trended towards better responses than to anti-PD-1 alone in both platelet-specific Garp KO mice (CR: 42% vs 23%) and their WT littermates (CR: 46% vs 13%), similar to observations in our previous experiments, although statistical significance was not reached. In contrast, Fig. 7c shows that anti-GARP:TGF- β 1 combined with anti-PD-1 showed the opposite trajectory to that seen in platelet-specific Garp KO mice or wild type mice. Anti-GARP:TGF- β 1 combined with anti-PD-1 in Treg-specific Garp KO mice trended towards inferior responses compared to anti-PD-1 alone (CR: 19% vs 28%), with the opposite trend observed in wild type litter mice (CR: 34% vs 20%).

Statistical significance was indeed not reached in these experiments, and we agree with the referee's suggestion to state this clearly. We have changed the paragraph accordingly, using a slightly more compact and direct wording:

“As shown in Fig. 7b, the proportions of complete responses to anti-GARP:TGF- β 1 + anti-PD-1 were superior to anti-PD-1 alone in platelet-specific *Garp* KO mice, as well as in their WT littermates (42% vs 23%, and 46% vs 13%, respectively). These results were in line with our previous experiments but did not reach statistical significance. In Treg-specific *Garp* KO mice, this difference was not observed (Fig. 7c), and if anything, the combination was modestly inferior to anti-PD-1 alone (not statistically significant).”

Lines 352-353

Our experiments in cell-specific Garp KO mice (<establish>) suggest that blocking the activity of TGF- β 1 emanating from GARP-expressing Tregs is required for anti-GARP:TGF- β 1 to exert antitumor activity.

We agree with the reviewer and have changed “establish” into “suggest”.

Line 353-354

Either delete the sentence: “Blocking the activity of TGF- β 1 emanating from GARP-expressing platelets is not necessary, and blocking that from other GARP-expressing cells, such as endothelial cells, is not sufficient”; or replace it with “Whether GARP-expressing platelets or GARP-expression from other cell types contributes to tumor rejection requires more extensive studies.”

We agree that our sentence was too assertive. We have added “They suggest also that...” before our sentence, which is now more accurate and addresses the reviewer’s comment. The paragraph reads:

“Our experiments in cell-specific *Garp* KO mice suggest that blocking the activity of TGF- β 1 emanating from GARP-expressing Tregs is required for anti-GARP:TGF- β 1 to exert anti-tumor activity. They suggest also that blocking the activity of TGF- β 1 emanating from GARP-expressing platelets or endothelial cells is neither necessary nor sufficient, although this requires further investigation.”

Reviewer #2 (Remarks to the Author):

The points raised in the previous round of review have been satisfactorily addressed.